# Dual-Channel Stopped-Flow Apparatus for Simultaneous Fluorescence, Anisotropy, and FRET Kinetic Data Acquisition for Binary and Ternary Biological Complexes

**DOI:** 10.3390/bios10110180

**Published:** 2020-11-19

**Authors:** Roberto F. Delgadillo, Katie A. Carnes, Nestor Valles-Villarreal, Omar Olmos, Kathia Zaleta-Rivera, Lawrence J. Parkhurst

**Affiliations:** 1Department of Chemistry, University of Nebraska-Lincoln, Lincoln, NE 68588-0304, USA; 2School of Engineering and Sciences, Tecnologico de Monterrey, Monterrey, NL 64849, Mexico; nestor.valles@tec.mx (N.V.-V.); oolmos@tec.mx (O.O.); 3BASF Enzymes LLC, 3550 John Hopkins Ct, San Diego, CA 92121, USA; 4GlaxoSmithKline, Medicinal Science and Technology, R&D, King of Prussia, PA 19406, USA; katie.x.carnes@gsk.com; 5Department of Bioengineering, University of California San Diego, San Diego, CA 92093, USA; kzaletar@eng.ucsd.edu

**Keywords:** stopped-flow, kinetics, fluorescence, anisotropy, polarizers, magic angle, FRET, trFRET, L-type instrumentation, steady-state anisotropy

## Abstract

The Stopped-Flow apparatus (SF) tracks molecular events by mixing the reactants in sub-millisecond regimes. The reaction of intrinsically or extrinsically labeled biomolecules can be monitored by recording the fluorescence, *F*(*t*), anisotropy, *r*(*t*), polarization, *p*(*t*), or FRET, *F*(*t*)*_FRET_*, traces at nanomolar concentrations. These kinetic measurements are critical to elucidate reaction mechanisms, structural information, and even thermodynamics. In a single detector SF, or L-configuration, the *r*(*t*), *p*(*t*), and *F*(*t*) traces are acquired by switching the orientation of the emission polarizer to collect the *I_VV_* and *I_VH_* signals however it requires two-shot experiments. In a two-detector SF, or T-configuration, these traces are collected in a single-shot experiment, but it increases the apparatus’ complexity and price. Herein, we present a single-detector dual-channel SF to obtain the *F*(*t*) and *r*(*t*) traces simultaneously, in which a photo-elastic modulator oscillates by 90° the excitation light plane at a 50 kHz frequency, and the emission signal is processed by a set of electronic filters that split it into the *r*(*t*) and *F*(*t*) analog signals that are digitized and stored into separated spreadsheets by a custom-tailored instrument control software. We evaluated the association kinetics of binary and ternary biological complexes acquired with our dual-channel SF and the traditional methods; such as a single polarizer at the magic angle to acquire *F*(*t*), a set of polarizers to track *F*(*t*), and *r*(*t*), and by energy transfer quenching, *F*(*t*)*_FRET_*. Our dual-channel SF economized labeled material and yielded rate constants in excellent agreement with the traditional methods.

## 1. Introduction

The fast-mixing apparatus was designed to track fast reactions in solution and it has played a fundamental role in chemistry, biochemistry, and molecular biology to reveal molecular interactions of proteins, DNA, RNA, enzymes, vitamins, or any other biomolecules that can be used as biosensors [1,2,3,4]. The Stopped-Flow (SF) origins began in 1923 with the continuous flow mixer of Hartridge and Roughton, but it required large-volume samples to have practical applications in molecular biology [5]. However, later modifications were introduced by Britton Chance to preserve material by accommodating small reaction volumes below ~500 μL [6]. In 1950, Quentin Gibson introduced a stopping syringe and a double mixer to properly invent the first SF [7]. Later, Robert Berger contributed with the high efficient ball mixer [8] that brought the mixing dead time in the sub-millisecond range [9]. Lastly, the last modern breakthrough was made by Gibson’s lab when the first computer-controlled SF system was introduced [7].

The natural fluorescence or the artificial labeling of biomolecules by fluorescent dyes allows one to tract association and dissociation reactions in the SF apparatus by following the fluorescence quenching or enhancement, *F(t)*, and the changes in the rotation of the dye-attached molecule by anisotropy, *r(t)*, and polarization, *p(t)* [10]. In addition, the SF apparatus can track Förster Resonance Energy Transfer, *F(t)_FRET_*, when an appropriated dye-pair is attached to each of the reactants or in the same molecule when conformational changes exist during binding [11]. In the *p(t)* case, it has been used to detect drugs of abuse [12,13,14], and the pesticide atrazine [15]. However, the *p(t)* signal is not a fundamental function and can lead to calculation errors; in contrast, the *r(t)* is a fundamental expression that is normalized to the total *F(t)* signal [16]. However, when *F(t)* varies due to changes in the dye quantum yield, *QY*, it is required to analyze the traces by following the product of *r(t)* and *F(t)* or *rF(t)* since it corrects the distortions and the kinetic traces can follow the exponential-decay behaviors [17].

Thus, when a single detection is used, the *r(t)* and *p(t)* sensing modalities are collected by alternating the position of the emission from vertical (*I_VV_* or *I_||_*) to horizontal (*I_VH_* or *I_⊥_)* with respect to the excitation polarizer, however, in a two-detector apparatus, the *I_VV_* and *I_VH_* are recorded simultaneously [18]. These two types of configurations are known as L-format and T-format SF, respectively. The former requires a double amount of the reactant solutions to acquire sequentially the *I_VV_* and *I_VH_* traces, making the L-format very wasteful of valuable material [19]. On the other hand, the T-format is more expensive but practical, and it has been employed with success to study the refolding and unfolding of several proteins and enzymes [17,20,21], and to measure the association and dissociation rate constants of several biomolecules [22,23,24,25,26,27]

Therefore, preserve valuable material, we have modified an L-type SF to collect simultaneously the *I_VV_* and *I_VH_* signals with a single experimental shot by using electronic filters and an instrument control system, which sorts out the *F(t)* and *r(t)* kinetic traces and stores them in spreadsheets for the corresponding analysis. Our dual-channel SF makes use of a photoelastic modulator (PEM) that vibrates at 50 kHz to modulate the vertical-polarized excitation laser-beam into circularly polarized light [28]. The PEM was first employed by J. Wampler and R. Desa for steady-state fluorimetry [29]. Later, Giblin-Parkhurst modified the PEM position for better signal gain to study the kinetic rates of the ribosome and the initiation factor 3 [1,30].

To corroborate the fidelity of the dual-channel SF, we tested the electronic circuit and the custom-tailored instrument control software by collecting the association traces of Oregon green^®^ biocytin (BcO) to avidin (AV) and compare them with the *F(t)* and *rF(t)* kinetic traces obtained with a set of two polarizers [19]. We further compare the dual-channel *F(t)* traces with the total fluorescein obtained with a single polarizer at the magic angle (54.7°) [18]. We continue testing the linearity of the instrument response for BcO-AV reactions at several concentrations and temperatures. Furthermore, we measured, by the dual-channel *F(t)* and *rF(t)* sensing modalities, the association kinetics of several dye-labeled DNA probes bearing the Adenovirus major late promoter, (TATAAAG, AdMLP), and the full-length and its core (N-terminal truncated) yeast Tata Binding Protein (yTBP and cTBP, respectively) [2,31,32,33]. We contrasted these dual-channel acquired traces with the *F(t)_FRET_* traces obtained with a double-labeled AdMLP functioning as a FRET probe. In addition, we monitor the ternary complex formation with preformed TBP-AdMLP complex and the Transcription Factor II A (TFIIA), which binds upstream of the TATA box and modulates conformation changes in the TFIID that enables promoter recognition and binding towards the formation of the Pre-Initiation Complex (PIC) which is required for the transcription initiation process [34,35,36,37,38,39].

## 2. Materials and Methods

### 2.1. Solution Conditions

All experiments were conducted in a buffered solution of 25 mM Tris, 100 mM KCl, 5 mM MgCl_2_, 1 mM CaCl_2_, and 2 mM DTT at pH. The SF reactions and steady-state experiments were acquired in a temperature range of 15 °C to 30 °C as indicated in each particular experiment. The temperature was controlled by a water bath (±0.02 °C) measured by a thermistor placed in the water bath of the temperature-controlled cuvette (Hellma Cells, Inc., Plainview, NY, USA).

### 2.2. Biological AV-BcO Materials

A summary of all association SF reactions is shown in Figure 1 and Table 1. The Oregon green^®^ 488 biocytin (BcO, lot 40300A) was purchased from Invitrogen (Eugene, OR, USA) and avidin (AV, CAS 1405-69-2, lot 608540) from Calbiochem (La Jolla, CA, USA). For the SF experimentation, the BcO concentration was 20 nM after mixing, and 200 nM, 260 nM, 520 nM, and 1040 nM for AV (in biotin site basis). The BcO and AV solutions were placed in syringe 1 and 2, respectively.

### 2.3. Oligonucleotide Probe Design

The dye-labeled top strands and the corresponding complements were synthesized by TriLink Biotechnologies, Inc. (San Diego, CA, USA). All strands were both HPLC and PAGE purified and the correct labeling was confirmed by comparing the dye’s peak absorbance ratios with respect to the 260 nm DNA absorbance [40]. The duplex DNA (ds) was prepared using a 10 × molar excess of complementary strands with a preincubation time of at least 20–30 min before reaction mixing. The dye-labeled top strand contained the TATAAAA box sequence from the Adenovirus Major Late Promoter (AdMLP) that is bound and bent by TBP [41]. Several dyes were used to label a 14-nucleotide top coding strand bearing the AdMLP (AdMLP_14ds_) and giving fluorescent probes of different sensitivity (Table 1, Figure 1). These dyes were attached to the single-strand oligomer probes by six-carbon linkers at the 3′ end with fluorescein (3′-Fl), and at the 5′ end either with x-rhodamine (5′-Xr) or TAMRA (N, N, N′, N′-tetramethyl-6-carboxyrhodamine, 5′-Ta). After complement binding, the duplexed probes were named AdMLP_14ds_*Fl, Xr*AdMLP_14ds_, and Ta*AdMLP_14ds_, respectively. A fourth single-labeled probe was a 31-nucleotide AdMLP sequence labeled with an internally labeled carboxy Fl attached to thymine (dT-Fl) by a nine-atom linker, which was named AdMLP_31ds_*Fl_int_, and was designed to accommodate simultaneously the TBP and TFIIA proteins (Table 1, Figure 1). A double-labeled probe, Xr*AdMLP_14ds_*Fl, was designed as a FRET probe to yield the *F(t)_FRET_* trace that monitors the donor quenching by energy transfer caused by DNA bending during the cTBP and yTBP binding. All the preformed duplex probes were placed in syringe 1 with at least 20–30 min incubation to reach the desired temperature under the water bath. The concentration of single and double labeled duplexes for all the SF experiments varied from 20 nM to 60 nM and to make comparative analysis between these different methodologies, it was needed to keep the protein/probe ratio constant.

### 2.4. Transcription Factor Proteins

The full-length Tata Binding Protein and the COOH terminal domain (or core domine) from *Saccharomyces cerevisiae* referred to as yTBP and cTBP, respectively; were expressed in *Escherichia coli*, purified, and concentrated in 25 mM HEPES-KOH, 20% glycerol, 1 mM EDTA, 1 mM DTT, and 300 mM KCl (pH 7.9) [2,42]. Both protein activities were determined by a titration protocol as described previously [2,3]; thus, all concentrations are reported for active proteins. The cTBP final concentrations, after mixing, were 98 nM for the dual-channel SF experiments, and 43 nM, 86 nM, and 166 nM for the FRET SF experiments. For the yTBP, the final concentrations, after mixing, were 210 nM, 220 nM, 420 nM, and 500 nM. All the cTBP and yTBP solutions were placed in syringe 2. The TFIIA binding SF kinetics were obtained with a preincubated TBP-AdMLP_31ds_ complex of 220 nM yTBP and 20 nM AdMLP_31ds_*Fl_int_ placed in syringe 1, and an 850 nM TFIIA solution placed in syringe 2, all final concentrations after mixing.

### 2.5. Dual-Channel SF

The association reactions were collected in our custom-made SF apparatus (Figure 2) [2,46]. The fluorescence emission was collected with a 520-nm interference filter (10BPF10-520 Oriel Corp., Stratford, CT, USA) at 20–40 nM, after reaction mixing, for the BcO, AdMLP_14ds_*Fl, Xr*AdMLP_14ds_*Fl, AdMLP_31ds_*Fl_int_ probes, and the respective complexes. For the Ta*AdMLP_14ds_ and Xr*AdMLP_14ds_ probes, the emission signal was acquired by the 580-nm (10BPF10-580) and 620-nm interference filters, respectively (10BPF10-620, Oriel Corp., Stratford, CT), at 40 nM and 60 nM, respectively, after mixing (Figure 1, Table 1). The SF G-factor values were calculated with the half-wave plate and an emission polarizer to obtain the *I_HV_*/*I_HH_* ratio, at 520 nm for the BcO, AdMLP_14ds_*Fl, and AdMLP_31ds_*Fl_int_ labeled probes and complexes, and at 580 nm and 620 nm for the Ta*AdMLP_14ds_ and Xr*AdMLP_14ds_ samples, respectively. The SF mixing dead time was 1 ms, and sufficient to detect a second-order rate constant of at least 1 × 10^9^ M^−1^s^−1^. The excitation light was provided by a Coherent Ar^+^ ion laser (Innova 70-4 Argon, Santa Clara, CA, USA) at 488 nm with an excitation power of 10–20 mW, for the BcO, AdMLP_14ds_*Fl, and AdMLP_31ds_*Fl_int_ probes, and respective protein-probe complexes. For the Ta*AdMLP_14ds_ and Xr*AdMLP_14ds_ samples, the excitation was set at 514.5 nm with an excitation power of 135 mW. The laser source was followed by a photo-elastic modulator (PEM-80; HINDS International, Inc., Portland, OR, USA) oriented 45° with respect to the electric (E) vector of the incident light, and the half-wave modulation was set at 50 kHz (Figure 2).

The dual-channel-SF filtering box consists of a 4-pole-pair digitally programmable band-pass (828BP), a passive high-pass filter with a set of capacitors, an active high-pass filter, a half-wave rectifier with negative output, and an integrator with capacitor selector (Figure 3). The demodulator circuitry provides a *V_DC_(t)* and a rectified *V_AC_(t)* voltage, yielding differing functions of the anisotropy as a function of time, *r(t)*, and the time-dependent total fluorescence, *F(t)*, by filtering out the desired frequency. The fluorescent signal is detected by R928 Photo-Multiplier Tube (PMT, Hamamatsu, Bridgewater, NJ, USA) having an emission spectral response of 185 to 900 nm, which is selected by an interference filter (520 nm, 580 or 620 nm, described above), and divided into two channels. The fluorescence light that passes the interference filter generates a voltage signal in the PMT that is fed as a “signal in” which is split into two signals. The first half-signal passes through an adjustable resister-capacitor filter (RC) to obtain a clean *V_DC_(t)* signal out. This RC filter has a variable capacitor set for the following time constants of 1 ms, 10 ms, 50 ms, 100 ms, and 500 ms and 1 s whose selection is equivalent to the time required for the capacitor voltage to decrease its initial voltage by up to 37%. The second split signal passes to an 828BP 4-pole-pair digitally programmable band-pass filter (Frequency Devices^TM^, Inc., Ottawa, IL, USA) to isolate the 50 kHz signal that is transmitted to an active high-pass filter, consisting of an OP37 High-speed Operational Amplifier, and an *RC* filter with a resistance selector (*RC* filter with an adjustable time constant). Subsequently, the signal is conducted to a half-wave rectifier with negative output, consisting of an OP271 operational amplifier, a resistor set, and two rectifier diodes. Finally, the signal is fed to an integrator with a capacitor selector, consisting of an OP271 operational amplifier, a resistor, and a capacitor selector that yield the rectified *V_AC_(t)* signal.

The effect of the PEM on the vertically polarized laser beam results in linear polarization, which is described by the Stokes’ (*Sp*) vector (Equation (1)), and for the sake of simplicity, we assume unit irradiance, with periodic retardation, *δ = δ*_0_ sin *ωt*, with scaling factor (*δ*_0_) and angular frequency (*ω*). At a π/2 scattering angle, the Stokes’ vector for fluorescence (*S_F_*) scattering is shown in Equation (2) [47], where the parameters *F*, *I*_0_, and *r* account for fluorescence, incident light, and the anisotropy function, in that order.
(1)$$S_{P}=\;\left(\begin{array}{cccc} 1 & 0 & 0 & 0\\ 0 & \cos\delta & 0 & \sin\delta \\ 0 & 0 & 1 & 0\\ 0 & -\sin\delta & 0 & \cos\delta \end{array}\right)\left(\begin{array}{c} 1\\ -1\\ 0\\ 0 \end{array}\right)=\left(\begin{array}{c} 1\\ -\cos\delta \\ 0\\ \sin\delta \end{array}\right)$$
(2)$$S_{F}=\frac{F}{6I_{0}}\;\left(\begin{array}{cccc} 4-r & -3r & 0 & 0\\ -3r & 3r & 0 & 0\\ 0 & 0 & 0 & 0\\ 0 & 0 & 0 & 0 \end{array}\right)\left(\begin{array}{c} 1\\ -\cos\delta \\ 0\\ \sin\delta \end{array}\right)=\frac{F}{6I_{0}}\left(\begin{array}{c} 4-r\left(1-3\;\cos\delta \right)\\ -3\;r\;\left(1+\cos\delta \right)\\ 0\\ 0 \end{array}\right)$$

The first term of the Stokes’ vector gives the total irradiance due to *F* scattering. Thus, the instantaneous response in a photomultiplier detector (*V*) is proportional to this term, the sensitivity (*S*), and the transducer gain (*κ*) of the detector (Equation (3)):(3)$$V=\frac{\kappa \;S\;F}{6I_{0}}\left[4-r\left(1-3\;\cos\delta \right)\right]$$

The averaged photomultiplier response over a PEM cycle becomes integrated by an RC filter of the demodulation circuitry resulting in direct current output, *V_DC_(t)* proportional to the *r(t)* function (Equations (4) and (5)), and a rectified *V_AC_(t)* signal (Equation (6)) that together yield the anisotropy *r(t)* (Equation (7)) and total fluorescence *F(t)* information (Equation (8)).
(4)$$V_{DC}=\frac{1}{t}\int_{t=0}^{t=k}Vdt=\frac{\kappa \;S\;F}{6I_{0}}\left\{4-r\left[1-3\;J_{0}(\delta_{0})\right]\right\}$$
(5)$$V_{DC}\sim F\cdot \left[1-0.47818\cdot \langle r\rangle \cdot \left(1+2.3806\cdot H\right)\right]$$
(6)$$V_{AC}\sim 1.5{\langle r\rangle }_{t}F_{t}\;$$
where *H = (1 − G)/(1 + G)* and *G = I_HV_/I_HH_* is the SF grating factor, the instrument sensitivity ratio towards vertically and horizontally polarized light, which was obtained using an emission polarizer perpendicular and parallel to the electric (*E*) vector, respectively. The *V_DC_(t)* and *V_AC_(t)* were baseline corrected to obtain the respective ratio as a function of time (*ρ_t_*) (Equation (7)) where the constant *A_Gain_* is the instrumental amplitude gain, which was calculated solving *ρ(t)* and *r(t)* at *t = ∞* which is equivalent to *r_ss_* of the complex. Finally, the total fluorescence signal *F(t)* is obtained after solving for the *r(t)* traces (Equation (8)).
(7)$$\rho \left(t\right)=\frac{V_{AC}\left(t\right)}{V_{DC}\left(t\right)}=\frac{1.5\cdot r\left(t\right)\cdot A_{Gain}}{1-0.47818\cdot r\left(t\right)\cdot \left(1+2.3806\cdot H\right)}$$
(8)$$F\left(t\right)=\frac{V_{DC}\left(t\right)}{1-0.47818\cdot r\left(t\right)\cdot \left(1+2.3806\cdot H\right)}$$

The data acquisition was managed by an instrument control software (8.0 LabVIEW™) to sort each channel into two separate spreadsheets (Figure 4). The converted digital data was collected at a rate of 1530, 3060, or 6120 data points per second for 655 μs, 327 μs, and 163 μs separation per data points whose signals from each channel were baseline subtracted. The two analog signals from the demodulator were digitized by PCI-5122 high-speed digitizers from National Instruments (Austin, TX, USA) with a 14-bit resolution and 100 MHz bandwidth, through channel 0 and 1, the instrumental control panel of which is shown in Figure 5.

### 2.6. Stopped-Flow Association Kinetics Acquired with Excitation and Emission Polarizers

The AV-BcO and yTBP-Ta*AdMLP_14ds_ association reactions, at 20 °C and 25 °C, respectively, were acquired by the traditional two-polarizer method to compare the resulted kinetic traces with those acquired by the dual-channel SF. The two-polarizer SF experiments were carried out with the same apparatus already described but the analog PMT voltage is fed directly to the digitizer and without the need of the dual-channel filtering box. The concentration of AV and BcO solutions was 20 nM and 200 nM, respectively; and the Ta*AdMLP_14ds_ and yTBP were 40 nM and 500 nM, respectively. The excitation wavelength was provided by the previously described Ar^+^ laser, and the fluorescence emission was collected through the 520 nm and 580 nm interference filters described above. The time-dependent *I_VV_(t)* and *I_VH_(t)* traces were collected individually in two experiments yielding *r(t)* according to Equation (9) [17], where *G* is the SF grating factor already described [19]. The *F(t)* in Equation (10) corresponded to the denominator of Equation (9):(9)$$r\left(t\right)=\frac{I_{VV}\left(t\right)-G*I_{VH}\left(t\right)}{I_{VV}\left(t\right)\;+\;2G*I_{VH}\left(t\right)}\;$$
(10)$$F\left(t\right)=I_{VV}\left(t\right)+2G*I_{VH}{\left(t\right)}_{\;}$$

### 2.7. Stopped-Flow Association Kinetics Collected at the Magic Angle

The *F(t)* association kinetics of BcO and AV reacting at concentrations of 20 nM and 200 nM, respectively, were also tracked with a single emission polarizer between the cuvette and the detector at the magic angle, *θ* = 54.74° [16,48]. The corresponding *F(t)* intensity is three times smaller than the intensity in Equation (10); however, it must yield equivalent eigenvalues to describe the kinetic trace. The excitation was set at 488 nm provided by the described Ar^+^ laser at 10–20 mW which polarization plane was orientated by a half-wave plate to be 90° with respect to the detector plane while the fluorescence emission was collected through the 520-nm interference filter previously described.
(11)$$F\left(t\right)=3\cdot F{\left(t\right)}_{\theta =54.74{}^{\circ}}\;$$

### 2.8. Steady-State Anisotropy, r_ss_

The *r_ss_* values of the free probes (e.g., BcO, Ta*AdMLP_14ds_, etc.) and protein-probe complex (e.g., AV-BcO, yTBP-Ta*AdMLP_14ds_, etc.) were collected with two polarizers or by the method of Giblin and Parkhurst (Equation (12)) [30]. The steady-state fluorescence signal was detected in a model A-1010 Alphascan fluorimeter (Photon Technologies, Inc.) equipped with an R928 PMT (Hamamatsu, Bridgewater, NJ, USA) with the emission-monochromator bandwidth set at 1 turn, and spectral response of 185 nm to 900 nm. For the polarizer method, the *r_ss_* was calculated according to Equation (9). The excitation was provided by a Xenon arc set at 480 nm for the BcO, AdMLP_14ds_*Fl, and AdMLP_31ds_*Fl probes and complex samples. The 535 nm and 560 nm excitation wavelengths were used for the Ta*AdMLP_14ds_ and Xr*AdMLP_14ds_ probes and complex samples, respectively. For the Giblin-Parkhurst method, the excitation was provided by the 488 nm line of the Coherent^®^ Ar^+^ ion laser already described. The 514.5 nm excitation line was used for the Ta*AdMLP_14ds_,Xr*AdMLP_14ds_, and its respective protein-probe complexes. The Giblin-Parkhurst method required a PEM set 45° with respect to the *E* vector of the laser beam and placed between the Coherent^®^ Ar^+^ ion laser and the sample compartment with a retardation level of 1.22·π. Two signals were obtained during 3–5 s with the power switch set at “on” and “off” position. The resulted signal was fitted to a straight line by the method of least squares to filter noise and photobleaching effects, yielding a ratio γ = on/off, that is used to calculate *r_ss_* according to Equation (12):(12)$$r_{ss}=\frac{4-4\gamma }{\gamma 3\left(2-3H\right)+\left(1-3A+3H+3HB\right)}$$
where *H = (1 − G)/(1 + G)*, and *G* is the fluorimeter grating factor, $A=\int_{a=1.55}^{b=1.62}\cos\left(1.22\pi \cdot \sin x\right)\cdot H/\pi dx$, and $B=\int_{a=1.55}^{b=1.62}\sin\;\left(1.22\pi \cdot \sin x\right)\cdot H/\pi dx$. The *r_ss_* values were calculated with at least six independent measurements at the reported temperatures in Table 2. The emission signals were selected by the fluorimeter’s monochromator at 520 nm for BcO, AdMLP_14ds_*Fl, and AdMLP_31ds_*Fl probes and respective complexes. The fluorescence emission at 580 nm and 620 nm were collected for the Ta*AdMLP_14ds_ and Xr*AdMLP_14ds_ samples. The fluorimeter *G*-factor was obtained from 500–700 nm by measuring the *I_HV_*/*I_HH_* ratio acquired by polarizers with 3–5 s scans carried out by triplicated with a step size of 5–10 nm and the emission monochromator bandwidth set at 1 turn. The fluorimeter *G*-factor calculations required an AdMLP_14ds_*Fl solution at a concentration of 0.5 μM, to yield the values in the 500–590 nm range. The fluorimeter *G*-factor in the 580–700 nm range was acquired by a solution of Xr*AdMLP_14ds_ at a concentration of 4.7 μM at excitations of 510 nm and 560 nm.

### 2.9. The F(t), r(t), and rF(t) Sensing Modalities Analysis

For the *F(t)* signal, the kinetic traces are dependent on the formation and disappearance of the involved fluorescence emitting species and their respective spectroscopy properties, according to Equation (13):(13)$$F\left(t\right)=\sum x_{i}\left(t\right)\varepsilon_{i(\lambda ex)}\cdot f_{i(\lambda ex)}\cdot QY_{i}=\;\sum x_{i}\left(t\right)\cdot {QY}_{i}^{eff}=\sum_{1}^{2}x_{i}\left(t\right)\cdot \Delta QY$$
where *x_i_* is the mole fraction of the fluorescence “*i*” species as a function of time (*t*), the *ε_i(λex)_* term is the molar extinction coefficient at the excitation wavelength, the *f_i_*_(λem)_ is the fluorescence spectral contour at the excitation wavelength, and the *QY_i_* is quantum yield of the specie “*i”* [17]. Since the excitation wavelength is constant (e.g., 488 nm or 514.5 nm) and the emission is collected by an interference filter (e.g., 520 nm, 580 nm, or 620 nm), the last three terms can be grouped and simplified as the effective quantum yield, *QY_i_^eff^ = ε_i_(λ_ex_)·f_i_(λ_ex_)·QY_i_*. When *i*= 2, the *QY_i_^eff^* is equal to the *QY* difference between the free and bound probe in the complex, as following: ${QY}_{i}^{eff}$= Δ*QY = QY_free_ − QY_bound_*.

The *r(t)* signal depends on *x_i_*, the specie’s intrinsic anisotropy, *r_i_*, and the respective *QY_i_* value, as shown in Equation (14), where the denominator corresponds to Equation (13).
(14)$$r\left(t\right)=\frac{\sum x_{i}\left(t\right)\cdot QY_{i}^{eff}\cdot r_{i}}{\sum x_{i}\left(t\right)\cdot QY_{i}^{eff}}=\frac{\sum_{1}^{2}x_{i}\left(t\right)\cdot \Delta QY\cdot r_{i}}{\sum_{1}^{2}x_{i}\left(t\right)\cdot \Delta QY}$$

In the case of large *QY* changes, the *r(t)* traces are distorted and do not follow the exponential decay models. However, the *r(t)* traces can be corrected by multiplying with the *F(t)* signals, resulting in a new function, *rF(t)*, as shown in Equation (15) [17]:

(15)$$rF\left(t\right)=\;\sum x_{i}\left(t\right)\cdot {QY}_{i}^{eff}\cdot r_{i}=\overset{2}{\underset{1}{\sum }}x_{i}\left(t\right)\cdot \Delta QY\cdot r_{i}$$

Since the three *F(t)*, *r(t*) and *rF(t)* sensing modalities have different amplitudes and values depending on the temperature and probe type (e.g., Fl, Xr, and Ta), it is better to normalize them to make a comparative analysis; which is denoted by a top bar, $\bar{r}\left(t\right)$, $\bar{F}\left(t\right)$ and $\bar{rF}\left(t\right)$, as shown in Equations (16)–(18), respectively:(16)$$\bar{r}\left(t\right)=\left(r\left(t\right)-r\left(0\right)\right)\;/\;\left(r\left(\infty \right)-r\left(0\right)\right)$$
(17)$$\bar{F}\left(t\right)=\left(F\left(t\right)-F\left(\infty \right)\right)\;/\;(F(\infty )-F\left(0\right))$$
(18)$$\bar{rF}\left(t\right)=\left(rF\left(t\right)-rF\left(0\right)\right)\;/\;\left(rF\left(\infty \right)-rF\left(0\right)\right)$$

The *F(t)* kinetic traces were fitted to mono- and bi-exponential decays, e.g., $\bar{F}\left(t\right)$
*= α × e(−λ × t) + C*, $\bar{F}\left(t\right)$
*= α*_1_ × *e(−λ*_1_ × *t) + α*_2_ × *e(−λ*_2_ × *t)* + C, respectively; where *α* is the pre-exponential and gives the relative phase contribution, where *α*_1_ + *α*_2_ = *1*, *λ* is the eigenvalue with units of reciprocal seconds (s^−1^), and *C* corresponds to a constant or baseline residual. In the case of other $\bar{r}\left(t\right)$ and $\bar{rF}\left(t\right)$ traces, the models are modified as following: $\bar{r}\left(t\right)$
*= 1 − α* × *e(−λ* × *t) + C* and $\bar{r}\left(t\right)$
*= 1 − [α*_1_
*× e(−λ*_1_
*× t) + α*_2_
*× e(−λ*_2_
*× t) + C]*, respectively.

### 2.10. Stopped-Flow Association Kinetics of TBP-AdMLP_14ds_ Collected by Energy Transfer

The binding association of the yTBP and cTBP proteins to AdMLP_14ds_ was tracked with the double-labeled Xr*AdMLP_14ds_*Fl probe. The Fl donor emission was acquired with the previously described 520-nm interference filter yielding the *F(t)_FRET_* traces whose intensities decrease as the acceptor (Xr) gets closer to the Fl dye caused by the DNA bending as TBP binds to the probe [2,3,49]. In other words, the *F(t)_FRET_* trace tracks the quenching of the donor as a function of the time due to the energy transfer towards the acceptor [2]. The *F(t)_FRET_* association traces were acquired in the previously described SF apparatus in which the analogous signal was fed directly to the digitizer. The resulted *F(t)_FRET_* traces were described by exponential decay models according to Equation (19). The excitation light was provided by the previously described Coherent Ar^+^ ion laser at 488 nm and 10–20 mW power.
(19)$$F{\left(t\right)}_{FRET}=\;\sum_{i=1}^{n}\alpha_{i}{*e}^{-\lambda_{i}*t}$$

### 2.11. Time-Resolved Energy Transfer, trFRET

To determine the DNA conformation changes caused by the TBP to two double-labeled AdMLP_14ds_ probes (Ta*AdMLP_14ds_*Fl and Xr*AdMLP_14ds_*Fl), we measured the inter-dye distance (*R*) according to the FRET rate of transfer (*k_t_*) which is inversely proportional to the reciprocal sixth power of interdye distance (*R^1/6^*) as shown in Equation (20), where τ_D_ is the donor lifetime reference of 4.1 ns ± 0.1 ns and the *R_0_* is the Förster distance of 61.8 Å ± 1.7 Å and 65.3 Å ± 0.3 Å for Xr*AdMLP_14ds_*Fl, and Ta*AdMLP_14ds_*Fl, respectively [40]. The solution concentrations of the single labeled donor (AdMLP_14ds_*Fl), and double-labeled Xr*AdMLP_14ds_*Fl and Ta*AdMLP_14ds_*Fl free probes were between 50–100 nM, and the respective complexes were formed by adding yTBP or cTBP to reach at least 1.5 μM to ensure >98% probe saturation, at 20 °C [3].

(20)$$k_{t}=\frac{1}{\tau_{D}}{\left(\frac{R_{0}}{R}\right)}^{6}$$

The time-resolved donor intensity in the nanosecond scale, *I_D_(t)*, of the single-labeled unbound AdMLP_14ds_*Fl and bound (yTBP-AdMLP_14ds_*Fl or cTBP-AdMLP_14ds_*Fl) complexes, were deconvoluted according to Equation (21), which yield *τ_D_* or <*τ_Di_*> for mono- or multiphasic decays, respectively. The excitation energy is lost by <*τ_Di_*> = 1/*k_F(D)_* = 1/(*k_D_*^0^ + Σ*k_i(D)_*), which is the reciprocal of the sum of the natural fluorescence rate (*k_D_*^0^ =1/*τ*^0^) and the inactivation pathways (Σ*k_i(D)_*). In addition, <*τ_Di_*> is equal to Σ*α_i_τ_i_*, the sum of the area under the curve for each of the *i*th lifetimes with its respective fractional contribution (*α_Di_*) so that Σ*α_Di_* = 1:(21)$$I_{Di}^{Exc/Emi}\left(t\right)=\sum_{i=1}^{n}\alpha_{Di}\cdot e^{-\left(\frac{t}{\langle \tau_{Di}\rangle }\right)\;}$$

The same time-resolved deconvolution model is used for the unbound and bound double-labeled duplexes, *I_D(A)_(t)*, where A is the acceptor, and the fluorescence donor emission is collected by the 520-nm interference filter already described:(22)$$I_{Di\left(A\right)}^{Exc/Emi}\left(t\right)=\sum_{i=1}^{n}\alpha_{Di(A)}\cdot e^{-\left(\frac{t}{\langle \tau_{Di(A)}\rangle }\right)\;}$$

The deconvolution yielded the lifetimes of single and double-labeled duplexes, the <*τ_Di_*> and <*τ_D(A)i_*> respectively, of free and TBP bound samples. To obtain the interdye *R* (Equation (23)), the <*τ_Di_*> is used as the donor reference lifetime for which the transfer rate constant (*k_t_*) is optimized as a function of a distance distribution, denoted as *P(R)* described by a mean inter-dye distance, $\bar{R}$, and a standard deviation, *σ* (Equation (24)). These two parameters of *P(R)* are optimized using nonlinear regression algorithms or a method of moments [2] to match the observed donor decay in the presence of the acceptor, <*τ_D(A)i_*>.
(23)$$I_{Di\left(A\right)}^{Exc/Emi}\left(t\right)=\int_{0}^{\infty }P\left(R\right)\sum_{i=1}^{n}\alpha_{Di}e^{\left[-\left(\tau_{Di}{}^{-1}+k_{t}\right)\cdot t\right]}dR$$
(24)$$P\left(R\right)=\frac{1}{\sigma \sqrt{2\pi }}\exp\left[-{\left(R-\bar{R}\right)}^{2}/\left(2\sigma^{2}\right)\right]$$

The corresponding DNA bend angle (*α*) was calculated according to Equation (25) in a simple rod model with two-kinks, based on the *R* for the free and bound double-labeled Xr*AdMLP_14ds_*Fl and Ta*AdMLP_14ds_*Fl probes [50].
(25)$$\frac{{\bar{R}}_{bound}}{{\bar{R}}_{unbound}}=\frac{\left({\bar{R}}_{free}-L_{2}\right)\cos\left(\propto /2\right)+\;L_{2}}{{\bar{R}}_{unbound}}$$


The raw *I_D_* and *I_D(A)_* decays were acquired using a dye-tunable laser pumped by an N_2_ LaserStrobe^®^ spectrofluorometer (PTI, Photon Technologies, Inc., Birmingham, NJ, USA) with a 10 Hz pulsed excitation set at 481 nm provided by PLD481 dye (Photon Technologies, Inc.). The decays were collected by filtering donor fluorescence emission through a 520-nm interference filter (10BPF10-520, Oriel Corp., Stratford, CT, USA) preceded by a liquid filter containing a 1 cm path length of 24.1 mM acetate-buffered dichromate (pH 4) to remove extra scattered light that may pass the interference filter. Three successive replicate decays were collected and immediately averaged to yield one sample decay having 120 points with 30 excitation pulses per point. Two instrument response functions (IRF) were collected with a glycogen solution for deconvolution purposes, and six sample curves were collected per set. At least four sets of six decays per set were collected per sample, which were deconvoluted in the nanosecond (ns) range to mono- bi- and tri-exponential decay models (Equations (21) and (22)) which were discriminated depending on the selection criteria of *χ^2^* between 0.9–1.1 [51], the residual correlation Durbin-Watson (*DW*) above 1.5 [52], and the runs test *Z* [53].

## 3. Results and Discussions

### 3.1. Steady-State Anisotropy

The fluorimeter *G*-factor values are used to correct for the detector-sensitivity deviations at the three emission wavelengths (e.g., 520 nm, 580 nm, and 620 nm) (Table 2, Appendix A). The fluorimeter *G*-factor at 520 nm was 0.7933 ± 0.0128 (Appendix A), which is the emission wavelength for the BcO, AdMLP_14ds_*Fl, and AdMLP_31ds_*Fl_int_ free probes and respective protein-bound complexes. The fluorimeter *G*-factor values, at 580 and 620 nm, were 0.6950 ± 0.0033 and 0.6388 ± 0.0134 at 620 nm, respectively (Appendix A), for the Ta*AdMLP_14ds_ and Xr*AdMLP_14ds_ labeled free and bound with TBP. Indeed, the free probes showed low *r_ss_* values that increased when complexed with the proteins, and the complex *r_ss_* values were used to solve for the association kinetic traces (Equation (7)) since they give the endpoint of the reaction at *r(t = ∞)*. To make sure it was calculated correctly, we added large excess of protein concentration until the *r_ss_* value remained unchanged, indicating that the reaction was driven to completion. Thus, in all the cases, the reactions were carried out at pseudo-first-order conditions with a protein excess of 10×, for at least 98% probe saturation. We observed an *r_ss_* temperature dependence in the BcO samples, from 10 °C, 15 °C, 20 °C, and 25 °C (Table 2), which was carefully determined to make a good kinetic comparative analysis.

### 3.2. Dual-Channel SF Validation by Polarizers and Magic Angle Methodologies

The association traces of BcO binding to AV were acquired with the *F(t)* and *r(t)* sensing modalities by the dual-channel SF methodology (Figure 6A–F) and the traditional two-polarizer method (Equations (9) and (10), Figure 6G,H). We also collected the *F(t)* with a single polarizer at the magic angle position (Equation (11)). For these three methodologies, the SF *G*-factor at 520 nm was 0.819 ± 0.015 (Appendix A). For the *F(t)* traces, the baseline-subtracted amplitudes were proportional to the change of the relative *QY* of the unbound and complex, Δ*F = (QY_unbound_ − QY_complex_)* (Table 2). Certainly, to make a comparison of the *F(t)* traces acquired at multiple excitation intensities and the *r(t)* at different temperatures, all traces were normalized according to Equation (16) to Equation (18), yielding the *r(t)*, $\bar{F}\left(t\right)$, and $\bar{rF}\left(t\right)$, respectively.

The $\bar{F}\left(t\right)$ traces were better described by a bi-exponential model; however, the second phase, *α*_2_ × *e(−λ*_2_*/t)*, corresponded to photobleaching as shown by traces acquired by discontinuous excitation (Figure 7A,B). The photobleaching eigenvalue ranged from λ_2_ = 0.01 s^−1^ to 0.02 s^−1^ depending on the laser intensity, and it is ignored in the data analysis. The kinetic traces were fitted to a reaction model of AV + BcO → AV−BcO, under pseudo-first-order condition (>10 × protein excess) after photobleaching was discarded. The biomolecular rate constant (*k_on_*) was obtained with the reaction eigenvalue, *λ*_1_ = *k’*[BcO], where *k’* = *k_on_*[AV]. The *k_on_* values calculated from the $\bar{F}\left(t\right)$ traces; acquired by the dual-channel, polarizers, and the single polarizer at the magic angle methodologies; were in excellent agreement with the overlapping errors (Table 3).

On the other hand, the Δ*F(t)* dropped by 25% and produced a small distortion in the *r(t)* traces; therefore, the *r(t)* needed to be analyzed as *rF(t),* as shown in Equation (15) [17]. As the AV-BcO complex is formed, the rotation of the probe decreased as shown by the $\bar{rF}\left(t\right)$ traces, which were collected only by two SF methodologies: polarizers and dual-channel SF (Figure 6F,H). The $\bar{rF}\left(t\right)$ traces acquired by these methodologies yielded equivalent eigenvalues, which were also in perfect agreement with the information obtained by the $\bar{F}\left(t\right)$ sensing modality. Thus, $\bar{F}\left(t\right)$ and $\bar{rF}\left(t\right)$ traces yielded *k_on_* overlapping values, which indeed validated the accuracy of the dual-channel SF methodology (Table 3, Figure 6).

### 3.3. Concentration and Temperature Dependence of AV-BcO Binding Association Acquired by Dual-Channel SF

After validation of the dual-channel methodology, we acquired the AV-BcO traces as a function of concentration and temperature. For the former case, the $\bar{F}\left(t\right)$ and $\bar{rF}\left(t\right)$ sensing modalities showed an increment in the reaction velocity when the protein concentration increased from 200 nM, 260 nM, 520 nM up to 1040 nM, for all the temperatures tested, from 10 °C to 25 °C. Consequently, when the concentration increased, the resulted fits yielded increasing λ values that were equivalent to both sensing modalities (Figure 7, Table 4). For instance, for the $\bar{F}\left(t\right)$ traces at 20 °C, the t_1/2_ were 584 ms ± 21 ms, 465 ms ± 8 ms, 228 ms ± 6 ms, and 112 ms ± 10 ms, and for the $\bar{rF}\left(t\right)$ traces, the t_1/2_ were 586 ms ± 11 ms, 449 ms ± 9 ms, 229 ms ± 2 ms, and 109 ms ± 1 ms, respectively. The λ values of $\bar{F}\left(t\right)$ and $\bar{rF}\left(t\right)$ traces at each concentration, resulted in *k_on_* values equivalent for both sensing modalities (Table 4).

The AV-BcO reaction speed also showed a temperature dependence from 10 °C to 25 °C from both $\bar{F}\left(t\right)$ and $\bar{rF}\left(t\right)$ sensing modalities. Thus, for the former traces, the t_1/2_ values were 1,102.0 ms ± 0.074 ms, 655 ms ± 0.010 ms, 465 ms ± 0.008 ms and 0.281 ms ± 0.002 s from 10 °C, 15 °C, 20 °C, and 25 °C. These t_1/2_ overlapped within the error of the $\bar{rF}\left(t\right)$ values of 1004.0 ms ± 0.050 ms, 670 ms ± 0.003 ms, 449 ms ± 0.004 ms and 0.280 ms ± 0.002 s, for the same order of the listed temperatures (Figure 7C,D). The *k_on_* values increased 47.9 % from 10 °C to 15 °C, 53.9 % from 15 °C to 20 °C, and 60.9 % from 20 °C to 25 °C, suggesting a strong temperature dependence in good accordance with ITC titrations [44].

### 3.4. The yTBP-AdMLP_14ds_ Association Traces Acquired by Dual-Channel and FRET SF Methodologies

To initiate gene transcription, the RNA polymerase II (RNAPII) needs the binding and bending of TBP to the TATA sequence (AdMLP) situated at the -31 nucleotide position of the first transcription codon [35]. Therefore, the yTBP-AdMLP complex formation is critical for gene expression [34] but it also requires several other proteins to form a multiprotein pre-initiation complex (PIC) to ensure the fidelity of transcription [32,36,39,45]. Here, we acquired the $\bar{F}\left(t\right)$ and $\bar{r}\left(t\right)$ traces of the yTBP (Figure 8, Appendix A) binding to the single labeled AdMLP_14ds_ probes labeled with tethered Ta, Xr, and Fl dyes and contrasted with *F(t)_FRET_* traces acquired with the double-labeled Xr*AdMLP_14ds_*Fl probe.

The *F(t)* traces of yTBP binding to the Ta*AdMLP_14ds_ and AdMLP_14ds_*Fl probes did not show any fluorescence change (Figure 8A). On the other hand, the $\bar{r}\left(t\right)$ sensing modality tracked very well the binding process (Figure 8B,C); therefore, the association kinetic can be studied with the $\bar{r}\left(t\right)$ traces and there is no need to obtain the $\bar{rF}\left(t\right)$ product as in the AV-BcO case (Figure 6 and Figure 7). To ensure that dual-channel traces were reliable for the yTBP-Ta*AdMLP_14ds_ complex formation, we compared them with the traces acquired by the polarizer methodology, at 25 °C (Figure 8B,C). Indeed, both methodologies yielded traces that resulted in overlapping λ values in the error (Figure 8B,C) that yielded the initial rate constant (*k_+_*_1_), at 25 °C, of the complex reaction mechanism previously elucidated by *F(t)_FRET_*, which consisted of six rate constants for a two-intermediates reaction [11].

To further evaluate the dual-channel *r(t)* traces, we compared yTBP binding, at 20 °C, with the single labeled AdMLP_14ds_*Fl (Figure 8D), Xr*AdMLP_14ds_ (Figure 8E), and the FRET Xr*AdMLP_14ds_*Fl probe (Figure 8F), at the same protein/probe ratio. The Δ*F(t)_FRET_* traces dropped 34.0% ± 2.0% (Equation (15)) for the yTBP-Xr*AdMLP_14ds_*Fl and the *F(t)_FRET_* traces were described by three eigenvalues (*λ*_1_, *λ*_2_, and *λ*_3_) whose faster component matched with the one observed with the *r(t)* traces (Figure 8D,E). The *r(t)* traces of yTBP-AdMLP_14ds_*Fl and yTBP-Ta*AdMLP_14ds_, at 20 °C, yielded *k_on_* values that overlapped in error between each other and as well with the *k_+_*_1_ acquired in this study by *F(t)_FRET_* and elsewhere [11]. To analyze conformational changes, we acquired the time-resolved FRET (trFRET) of the free single labeled AdMLP_14ds_*Fl and bound to yTBP and cTBP. We also obtained trFRET of the unbound double-labeled Xr*AdMLP_14ds_*Fl and Ta*AdMLP_14ds_*Fl, and the respective yTBP and cTBP complex (Table 5) to obtain the $\bar{R}$ and *σ* parameters for free duplexes and complexes (Figure 9A,B). The trFRET lifetime in ns, Σ*α_i_τ_i_* or *τ_D(A)_*, of the free and yTBP bound probe, Xr*AdMLP_14ds_*Fl and yTBP-Xr*AdMLP_14ds_*Fl were 1.575 ns ± 0.066 ns and 1.042 ns ± 0.040 ns, respectively. The Δ*t_D(A)_* between the free Xr*AdMLP_14ds_*Fl and bound to the yTBP-Xr*AdMLP_14ds_*Fl complex was 33.8% ± 2.0 % and it was equivalent to the observed Δ*F(t)_FRET_* in the association traces. Likewise, the Δ*t_D(A)_* when Ta was used as acceptor was 33.5% ± 1.7% (Table 5). The bend angles *α* were equivalent for both yTBP-Ta*AdMLP_14ds_*Fl and yTBP-Xr*AdMLP_14ds_*Fl complexes with values of 79.8° and 79.2°, respectively (Figure 9C).

### 3.5. Core TBP-AdMLP_14ds_ Association Kinetics Acquired by Dual-Channel and FRET SF Methodologies

We also contrasted the $\bar{r}\left(t\right)$ dual-channel association trace of cTBP and AdMLP acquired with the single labeled Xr*AdMLP_14ds_ probe (Figure 10A, purple line, Appendix A) and the *F(t)_FRET_* trace collected with the double-labeled Xr*AdMLP_14ds_*Fl probe (Figure 10A, yellow line). The association reaction reactions were collected under discontinuous excitation to eliminate the photobleaching effect and the respective fit passed through the traces (solid and slashed lines, Figure 10A). The cTBP has the N-terminal domain truncated in comparison with the full-length yTBP and as previously described for full-length yTBP-AdMLP, the $\bar{F}\left(t\right)$ trace of cTBP-Xr*AdMLP_14ds_*Fl was not sensitive to the protein binding but the $\bar{r}\left(t\right)$ trace tracked the complex formation by increasing its value as the dye rotation decreased. For the FRET probe, the $\bar{F}{\left(t\right)}_{FRET}$ trace decreased as the cTBP bent the probe, resulting in a decrement in the fluorescence by energy transfer. To visualize the comparison of these two traces, we inverted the $\bar{r}\left(t\right)$ trace and yielded just over the $\bar{F}{\left(t\right)}_{FRET}$ whose fitting lines overlap perfectly (Figure 10B).

The *F(t)_FRET_* traces of the cTBP-AdMLP reactions showed concentration dependence when the protein concentration increased from 43 nM, 86 nM, and 165 nM (Figure 10 C) while the concentration of Xr*AdMLP_14ds_*Fl, was constant at 20 nM, at 20 °C. Furthermore, there was a temperature dependence as the reaction speed increased from 15 °C, up to 25 °C, at 86 nM cTBP and 20 nM Xr*AdMLP_14ds_*Fl (Figure 10D). These observed reactions were completed at 49.0% ± 1.0% whose fits showed three exponential phases (black lines). In addition, we measured the trFRET lifetimes of the unbound Xr*AdMLP_14ds_*Fl and Ta*AdMLP_14ds_*Fl and complexed by cTBP (Table 5) whose corresponding changes were 48.9% ± 3.0% and 49.5% ± 2.0%, respectively, perfectly matching the *F(t)_FRET_* association change. The $\bar{R}$ and *σ* were obtained for the free probes Xr*AdMLP_14ds_*Fl and Ta*AdMLP_14ds_*Fl and the respective complexes formed by the cTBP (Figure 9B). The cTBP bend angle was larger than *α* produced by the full-length yTBP (Figure 9C) since the N-terminal domain has a regulatory activity [2]. Both protein complexes formed with Xr*AdMLP_14ds_*Fl and Ta*AdMLP_14ds_*Fl yielded equivalent bend angles showing that acceptor Ta and Xr dyes are excellent FRET biosensors and yielded the same results.

### 3.6. Ternary Association Kinetics of TFIIA and the Binary yTBP-AdMLP_31ds_ Acquired by Dual-Channel SF

The function of TFIIA in the PIC complex is to shift the equilibrium towards a rearranged more stable state, as shown by 3D cryo-EM reconstructions [54]. To test out the dual-channel SF ability to measure large complexes, we used a longer 31mer internally-labeled with fluorescein (AdMLP_31ds_*Fl_int_) to allow space for the TFIIA binding at the 5′ upstream of the TATA sequence [39,41,55]. First, we validated the yTBP binding to the longer AdMLP_31ds_*Fl_int_, and in contrast to the 3′Fl in the shorter AdMLP_14ds_ probes, the Fl_int_ was sensitive to yTBP binding since the *QY* dropped 15.5% (Table 2), and consequently the binding was followed as well by the *F(t)* sensing modality (Figure 11A). Interestingly, the *QY* increased after TFIIA binding (Table 2). To compensate for *QY* changes, the *r(t)* was multiplied by *F(t)* to analyze the $\bar{rF}\left(t\right)$ sensing modality, at 20 °C and 30 °C (Figure 11B), thus the resulting *k_on_* overlapped with the reported *k_+_*_1_ values at the same temperatures indicating that longer probe is functional [11] and in excellent accord with the shorter single labeled and double-labeled probes (Figure 8).

After confirming that the longer probe is functional, we pre-formed the binary yTBP-AdMLP_31ds_*Fl_int_ complex and reacted with TFIIA, at 15 °C, 17 °C, and 25 °C, which was tracked by the *rF(t)* sensing modality (Figure 11C). The *k_on_* values of TFIIA binding to the binary complex, at 15 °C 17 C, and 25 °C, were 1.45 (±0.3) × 10^6^ M^−1^s^−1^ and 4.68 (±1.59) × 10^6^ M^−1^s^−1^. There is a reported dissociation rate constant of the ternary TFIIA-TBP-TATA of 7.1 × 10^−4^ s^−1^ at room temperature [38], which let us estimate the *K_D_* values between 2.1 nM and 6.6 nM for yTFIIA and the binary yTBP-TATA complex, between the 15 °C and 25 °C range.

## 4. Conclusions

We evaluated the electronic filters and custom-tailored instrument control system of a new dual-channel SF apparatus that allows simultaneous acquisition of the *F(t)* and *r(t)* with an optical train in L-type configuration [18,19]. In our dual-channel SF apparatus, we economized by a factor of two the biomolecule solution consumption in comparison to the polarizer SF, allowing us to preserve precious labeled biological material such as proteins, DNA, RNA, ligands, and other labeled biomolecules [19]. To validate our dual-channel SF, we tracked multiple association reactions at diverse conditions and contrasted the resulted *k_on_* values with those acquired by other methodologies with multiple sensing modalities, such as *r(t)*, *F(t), rF(t),* and *F(t)_FRET_*.

In conclusion, the dual-channel SF has functional and robust electronic filtering since the tested circuit separates the *V_DC_* and *V_AC_* signals and stores them in separate spreadsheets, acquiring the *r(t)* and *F(t)* information in one single shot. The calculated *k_on_* values acquired by the dual-channel SF traces showed ~50% lower noise levels, as shown in the errors reported in Table 3. Our new setting was reliable and allowed the association traces of relevant biological complexes by monitoring the *F(t)*, *r(t)*, and *rF(t)* sensing modalities. Our work describes new hardware to collect kinetic data at different concentrations and temperatures to elucidate reaction mechanisms [56] and thermodynamic information according to the Arrhenius equation [57,58] and Eyring relationship [59,60], which is relevant for protein-protein, protein-DNA, or protein-drug studies [61,62].

## Figures and Tables

**Figure 1 biosensors-10-00180-f001:**
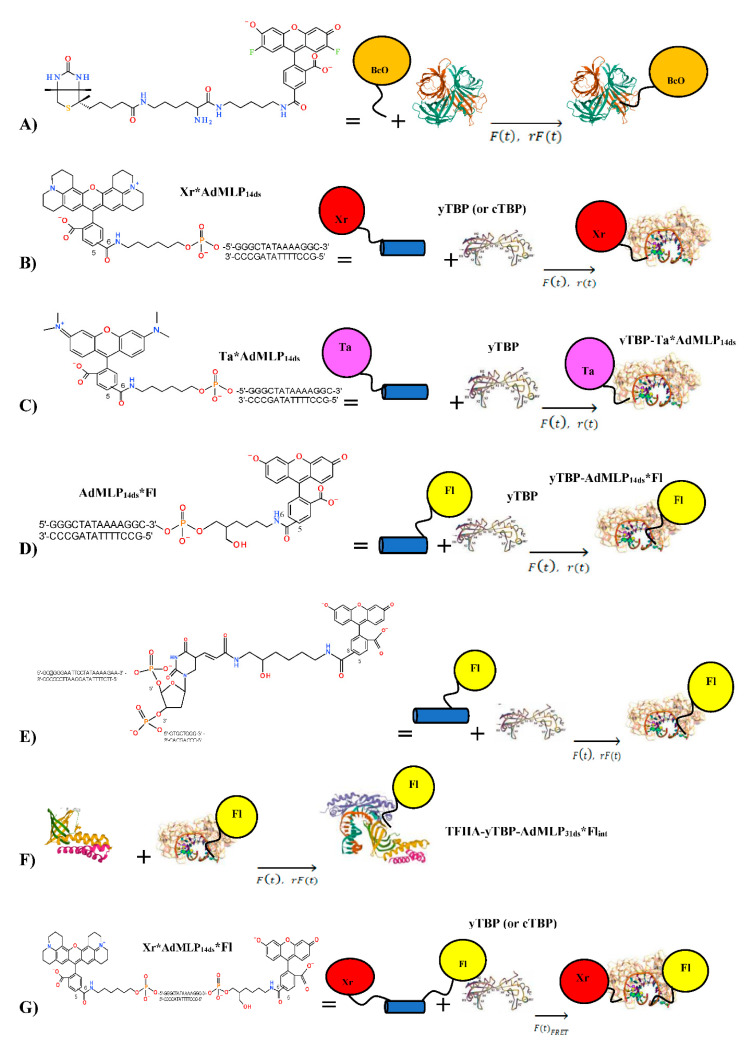
Stopped-flow association reactions monitored by *r(t)*, *F(t)*, *rF(t)*, *and F(t)_FRET_* sensing modalities. (**A**) The Oregon Green ^®^ Biocytin (BcO) is attached through 18 non-hydrogen atoms as a spacer between the carboxy-dye and biotin ring structure, which binds to one of the four sites of avidin (AV) [43] under pseudo-first-order condition (or > 10× binding site excess) [44]. (**B**–**G**) The Adenovirus Major Late promoter duplexed probes (AdMLP_ds_, Table 1) are bound and bend by TATA-binding protein (TBP) [2,31]. To track the TBP-AdMLP association reactions, several duplex probes were used with three different dyes and two TATA bearing sequences of 14-nucleotide (AdMLP_14ds_) and 31-nucleotide (AdMLP_31ds_) oligomer lengths. (**B**) The AdMLP_14ds_ probes were labeled at the 5′ end by x-rhodamine (Xr*AdMLP_14ds_), (**C**) TAMRA (Ta*AdMLP_14ds_), and (**D**) fluorescein at the 3′ end (AdMLP_14ds_*Fl). These 5′-Xr, 5-Ta, and 3′-Fl dyes were attached by six-carbon linkers. (**E**) The yTBP binding was also tracked with a fourth single labeled probe, a longer AdMLP_31ds_ sequence, labeled internally by fluorescein (AdMLP_31ds_*Fl_int_) attached by a nine-atom linker to the methyl group of the d-thymine (dT*Fl_int_) at position 23, and it was designed to accommodate at the same time the TBP and the TFIIA. (**F**) The extra space at the 5′ end allows us to obtain the TFIIA association rate constant when binding to a preformed yTBP-AdMLP_31ds_ binary complex thus forming the ternary TFIIA-yTBP-AdMLP_31ds_*Fl_int_ complex [45]. (**G**) The yTBP was also reacted with the double-labeled AdMLP duplex (Xr*AdMLP_14ds_*Fl) to acquire the *F(t)_FRET_* traces as the inter-dye probe distance was decreased by the protein bending thus quenching the donor fluorescence signal.

**Figure 2 biosensors-10-00180-f002:**
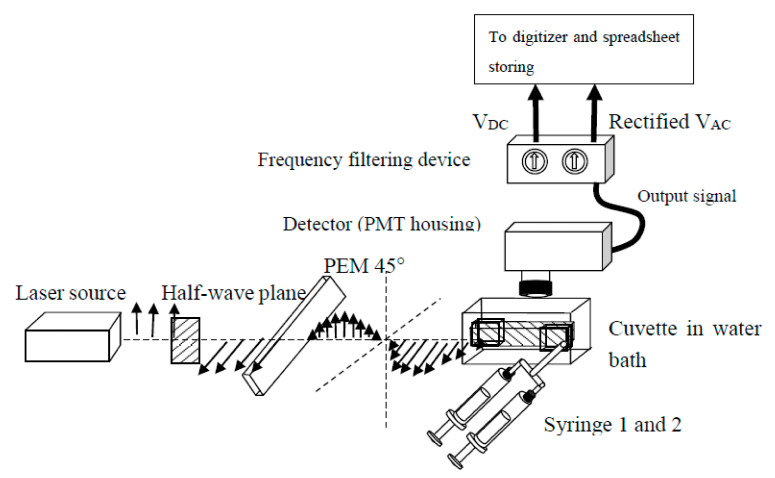
The modular dual-channel Stopped-Flow apparatus is equipped with a frequency rejection circuit for anisotropy, *r(t)*, and fluorescence, *F(t)*, detection. The optical train consists of the Coherent Ar^+^ ion laser (Innova 70-4 Argon, Santa Clara, CA, USA) source, a half-wave plate, the PEM at 45° of the vertical electric vector, and the detector placed in an L-type configuration. The half-wave plate changes by 90° laser excitation plane just before being modulated by the PEM. The detector housing can accommodate interchangeable interference filters (e.g., 520 nm, 580 nm, and 620 nm) to collect the fluorescence emission of the diverse dyes. The fluorescence emission is split into two signals, *V_DC_(t)* and rectified *V_AC_(t)*, by the electronic filters to be later digitized and stored in spreadsheets to yield *r(t)* and *F(t)* according to Equation (7) and Equation (8), respectively. The temperature of the cuvette and syringes is controlled by a water bath from 10 °C to 30 °C. The syringes are simultaneously pushed by an air-controlled piston to deliver 150 μL solution each to be ball mixed with a 1 ms death time.

**Figure 3 biosensors-10-00180-f003:**
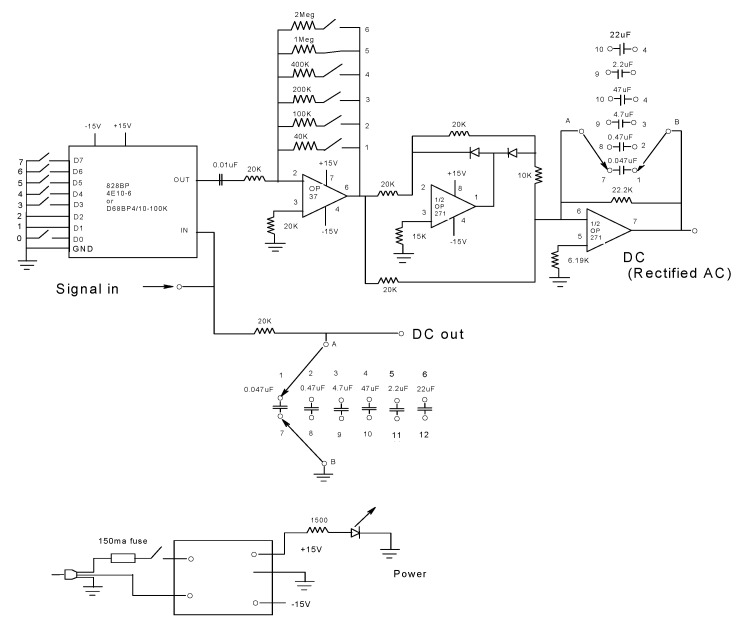
Circuit diagram for the dual-channel SF filtering box. The frequency selection circuitry filtered out the *r(t)*, and *F(t)* traces by getting the ratio of *V_AC_(t)*/*V_DC_(t)*, according to Equation (7) and Equation (8), respectively. The fluorescence emission that passes the interference filter (e.g., 520 nm, 580 nm, and 620 nm) produces a PMT analog signal (signal in) that is split into two, the first half goes through an adjustable resister-capacitor filter (*RC*) to obtain a clean *V_DC_(t)* signal out. This *RC* filter has a variable capacitor set for the following time constants of 1 ms, 10 ms, 50 ms, 100 ms, and 500 ms and 1 sec for the dials starting from 1 to 6. The second split signal passes through an 828BP 4-pole-pair digitally programmable band-pass filter (Frequency Devices ^TM^, Inc.) to isolate the 50 kHz signal, which is then fed to an active high-pass filter and later to a half-wave rectifier with negative output. Finally, the signal is fed to an integrator with a capacitor selector to yield the *V_AC_(t)*. The configuration of these electronic elements is powered by a 12V power supply. Further details are described in the text.

**Figure 4 biosensors-10-00180-f004:**
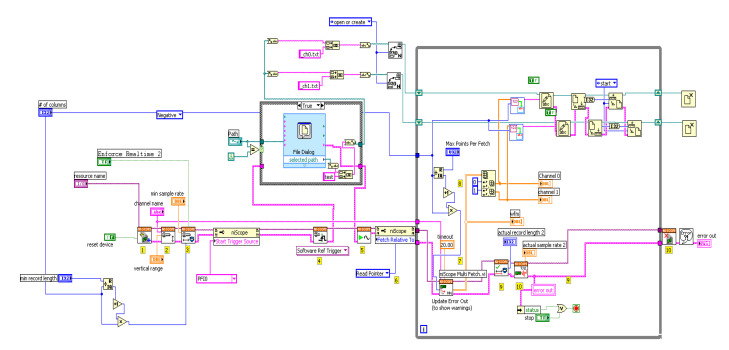
Schematic representation of the SF dual-channel LabView^®^ instrument control software. The instrument virtual software prepares data acquisition to a sample rate of 1536, 3072, and 6144 data points per second and creates two text files to store the output *V_DC_(t)* and *V_AC_(t)* signals in channel 0 and 1, respectively. The virtual range corresponds to the voltage variation in the detector from -5V up to +5V. The collection protocol can be started manually or by a triggering signal (PF10) when the stopping syringe mechanism is closed (Figure 5). The waiting period for receiving the triggering signal is set at 20 s. When saturation is reached at 5V, the data acquisition is truncated. The background is collected with the buffer solution to eliminate it from the *r(t)* and *F(t)* traces.

**Figure 5 biosensors-10-00180-f005:**
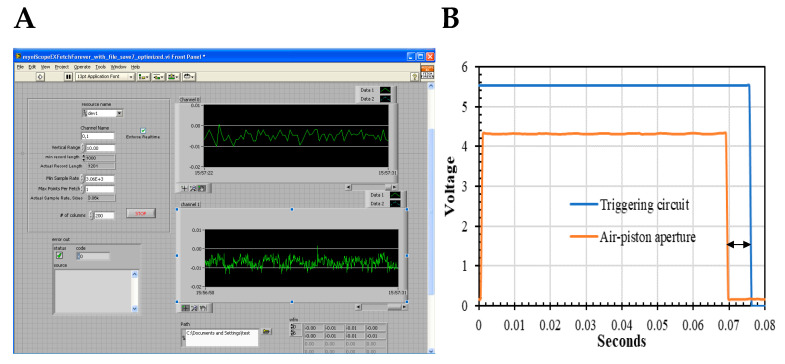
National Instrument front-screen interface for the dual-channel SF. (**A**) Channel 0 and channel 1 panels showing the *V_DC_(t)* and *V_AC_(t)* split signals, respectively, whose vertical range is 10 V from −5 V to +5 V. The data speed acquisition was set at 1530, 3060, or 6120 data points per second in the “Min Sample Rate” input, which can be stored horizontally in columns, and are set in the “# of columns” input. (**B**) The triggering signal is received in the PFI0 input (Figure 4), which has a waiting time window of 20 s to initiate data collection. The aperture of the air piston pushes the two solution syringes in a 67 ms ± 1 ms window (orange line). The data-acquisition triggering is started by the stopping syringe mechanism that has a feedback signal (blue line) with a deadtime of 7 ms ± 1 ms with respect to the air-piston closure (double-arrow separation). In addition, data acquisition can be pre-triggered manually resulting in a death-time of ~1 ms, which depends on the ball mixing effectiveness.

**Figure 6 biosensors-10-00180-f006:**
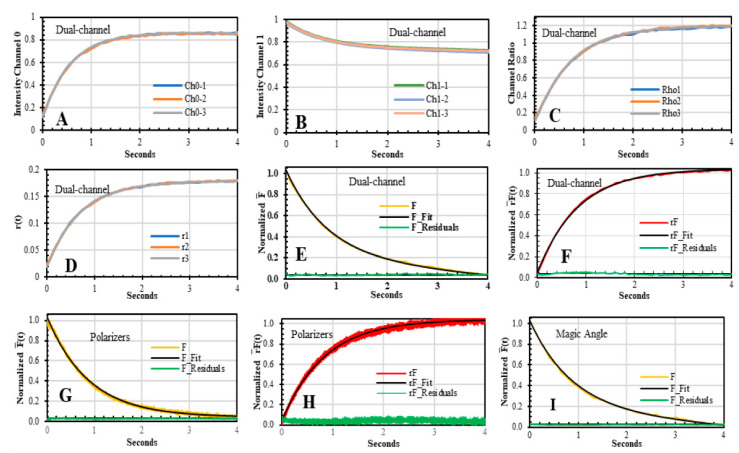
Stopped-flow kinetic traces acquired by dual-channel (**A**–**F**), polarizers (**G**,**H**), and magic-angle (**I**) methodologies for the binding association of BcO (20 nM) and AV (200 nM) at 20 °C. The *F(t)* and *rF(t)* sensing modalities were employed to track the AV-BcO complex formation. For the former trace, the Δ*F* decreased as the BcO fluorescence is quenched when the complex is formed and it is equivalent to Δ*QY* (Table 2), and in the case of the *rF(t)*, the trace value increased as the probe binds to the protein decreasing its rotation. (**A**,**B**) The dual-channel *V_AC_(t)* and *V_DC_(t)* traces were collected in channels 0 and 1, respectively. (**C**,**D**) The endpoint ratio of channel 0 and channel 1, *ρ(∞)* = 1.2, and the *r_ss_* of the AV-BcO complex are used to solve for the *r(t)* traces according to Equation (7) (Table 2) to later obtain the $\bar{F}\left(t\right)$ (Equation (8)), and subsequently the $\bar{rF}\left(t\right)$ traces. (**E**) The $\bar{F}\left(t\right)$ traces were solved with Equation (8) and the respective fitted curves (black line, and residual in green) yielded two eigenvalues of *λ*_1_= 1.186 s^−1^ ± 0.046 s^−1^ and *λ*_2_ = 0.02 ± 0.01 s^−1^ (Table 3). The latter *λ* corresponds to photobleaching and is neglected from further analysis. (**F**) The $\bar{rF}$*(t)* fitting parameters yielded an *λ* = 1.183 s^−1^ ± 0.023 s^−1^ as the probe’s rotation decreases in the newly formed complex. (**G**,**H**) The $\bar{F}\left(t\right)$ and $\bar{rF}\left(t\right)$ association traces acquired by polarizers yielded mono-exponential decays with *λ*_1_ = 1.198 s^−1^ ± 0.039 s^−1^, and *λ*_1_ = 1.187s^−1^ ± 0.083 s^−1^, respectively. The photobleaching was *λ*_2_ = 0.02 s^−1^ ± 0.01 s^−1^. (**I**) The $\bar{F}\left(t\right)$ traces collected by a single polarizer at the magic angle (54.7°) yielded *λ*_1_ = 1.199 s^−1^ ± 0.099 s^−1^ and the photobleaching decay was *λ*_2_ = 0.01 s^−1^ ± 0.01 s^−1^.

**Figure 7 biosensors-10-00180-f007:**
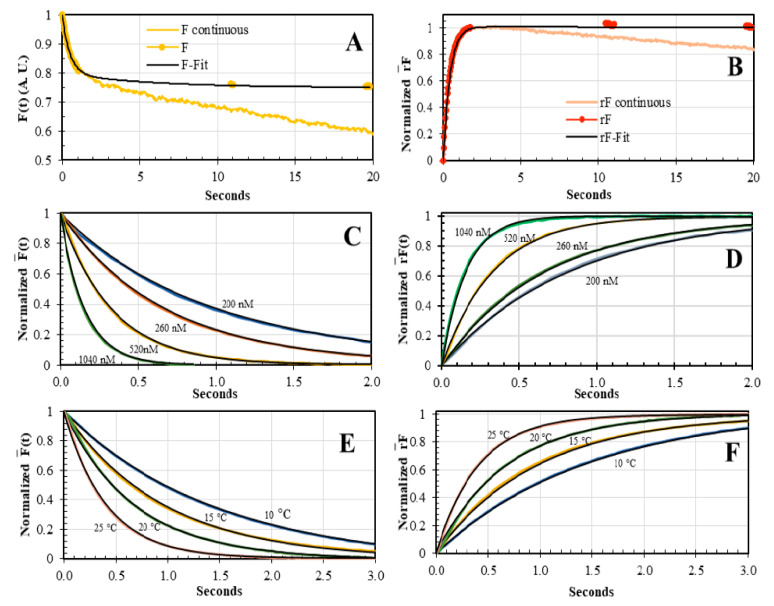
Concentration and temperature dependence of the AV-BcO association kinetics. (**A**,**B**) The *F(t)* and $\bar{rF}\left(t\right)$ sensing modalities were collected with the dual-channel SF under continuous (pale line) and discontinuous (dotted line) laser excitation at 488 nm. The *F(t)* trace, under continuous excitation, showed a second eigenvalue (*λ*_2_) that ranged from 0.02 s^−1^ to 0.01 s^−1^ caused by photobleaching. In contrast, when the excitation was blocked, the intensity stayed constant after the reaction was completed, so that, the intensity decreased by 25.3 ± 2.2% and it was proportional to a change in the *QY* of 0.91 ± 0.01 for the free BcO probe and the 0.68 ± 0.02 of the AV-BcO complex (Table 2). (**C**,**D**) The concentration-dependence of the binding reaction of BcO (20 nM) and AV (200 nM, 260 nM, 520 nM and 1040 nM), at 20 °C, for both $\bar{F}\left(t\right)$ and $\bar{rF}\left(t\right)$ sensing modalities, respectively. (**E**,**F**) The temperature-dependence in the binding reactions of BcO (20 nM) and AV (260 nM) at 10 °C, 15 °C, 20 °C and 25 °C, were tracked by $\bar{F}\left(t\right)$ and $\bar{rF}\left(t\right)$ modalities. The black lines corresponded to the fitted curves that yielded the λ and *k_on_* values, shown in Table 4.

**Figure 8 biosensors-10-00180-f008:**
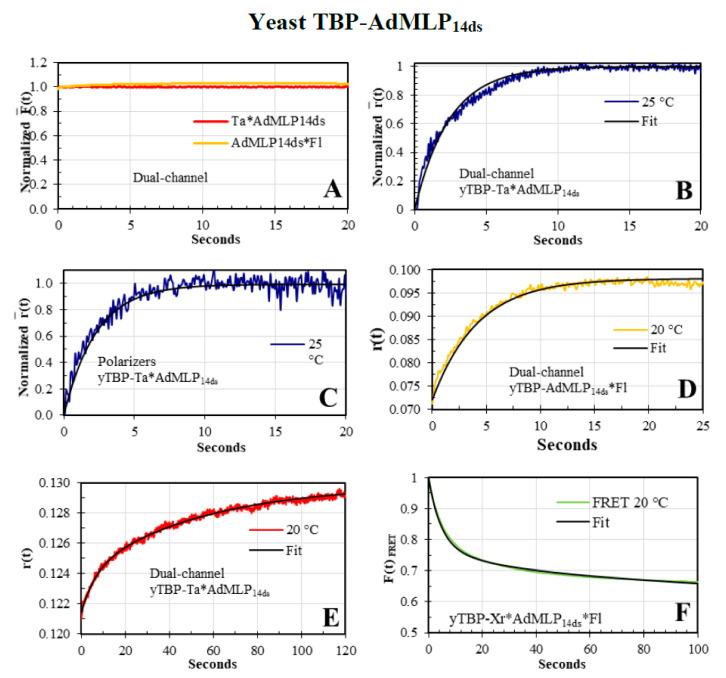
Yeast TBP-AdMLP_14ds_ association kinetics. (**A**) The dual-channel $\bar{F}\left(t\right)$ traces of yTBP (500 nM)-Ta*AdMLP_14ds_ (40 nM, red), and yTBP (420 nM)-AdMLP_14ds_*Fl (40 nM, orange line) were not sensible to the binding process; in contrast, the *r(t)* traces were sensible to the complex formation. (**B**,**C**) The $\bar{r}\left(t\right)$ association reaction of yTBP (500 nM)-Ta*AdMLP_14ds_ (40 nM), at 25 °C, were acquired by the dual-channel and polarizer SF methodologies whose fitted traces (black line) were mono-exponentials, *1 – α × e(−λ × t*), with *λ*= 0.384 s^−1^ (±0.020 s^−1^) and *λ* = 0.436 s^−1^ (±0.035 s^−1^), respectively. The dual-channel and polarizer *r(t)* traces resulted in *k_on_* values of 1.54 (±0.08) × 10^6^ M^−1^s^−1^ and 1.74 (±0.14) × 10^6^ M^−1^s^−1^, respectively; and were in the error range of first rate constant (*k_+_*_1_) of 1.59 [0.03−0.07] × 10^6^ M^−1^s^−1^, reported by FRET analysis, at 25 °C [11]. (**D**) At 20 °C, the *r(t)* dual-channel SF association reaction of yTBP (420 nM)-AdMLP_14ds_*Fl (20 nM) was also mono-exponential (black line) with a *λ* = 0.247 s^−1^ (±0.025) s^−1^, which yielded a *k_on_* of 5.87 (±0.60) × 10^5^ M^−1^s^−1^. (**E**) The 5′-Ta single labeled probe was used to acquire the *r(t)* dual-channel SF association reaction of yTBP (440 nM)-Ta*AdMLP_14ds_ (40 nM), at 20 °C. The trace was fitted to a bi-exponential model (black line) with the following parameters: *α*_1_ = 97.39% (±0.20%), *λ*_1_ = 0.1276 s^−1^ (±0.0256 s^−1^), *α*_2_ = 2.61% (±0.02)%, *λ*_2_ = 0.0181 s^−1^ (±0.0200 s^−1^). The faster phase yielded in a *k_on_* of 5.80 (± 1.16) × 10^5^ M^−1^s^−1^ that is excellent agreement with the 5.80 (± 0.26) × 10^5^ M^−1^s^−1^ reported, at 20 °C [11]. (**F**) The FRET SF association reaction of yTBP (220 nM)-Xr*AdMLP_14ds_*Fl (20 nM), at 20 °C, showed a triphasic exponential-decay model: *α*_1_ = 30.9% (±0.5%), *λ*_1_ = 0.22 s^−1^ (± 0.01 s^−1^), *α*_2_ = 11.9% (±2.1%), *λ*_2_ = 0.040 s^−1^ (±0.013 s^−1^) and *α*_3_ = 58.2% (±0.9%), *λ*_3_ = 0.0012 s^−1^ (±0.0002 s^−1^). The fast *λ*_1_ yielded a *k_on_*= 5.50 (±0.25) × 10^5^ M^−1^s^−1^ which is in excellent agreement with the *k_+_*_1_= 5.80 (±0.26) × 10^5^ M^−1^s^−1^ reported at 20 °C [11].

**Figure 9 biosensors-10-00180-f009:**
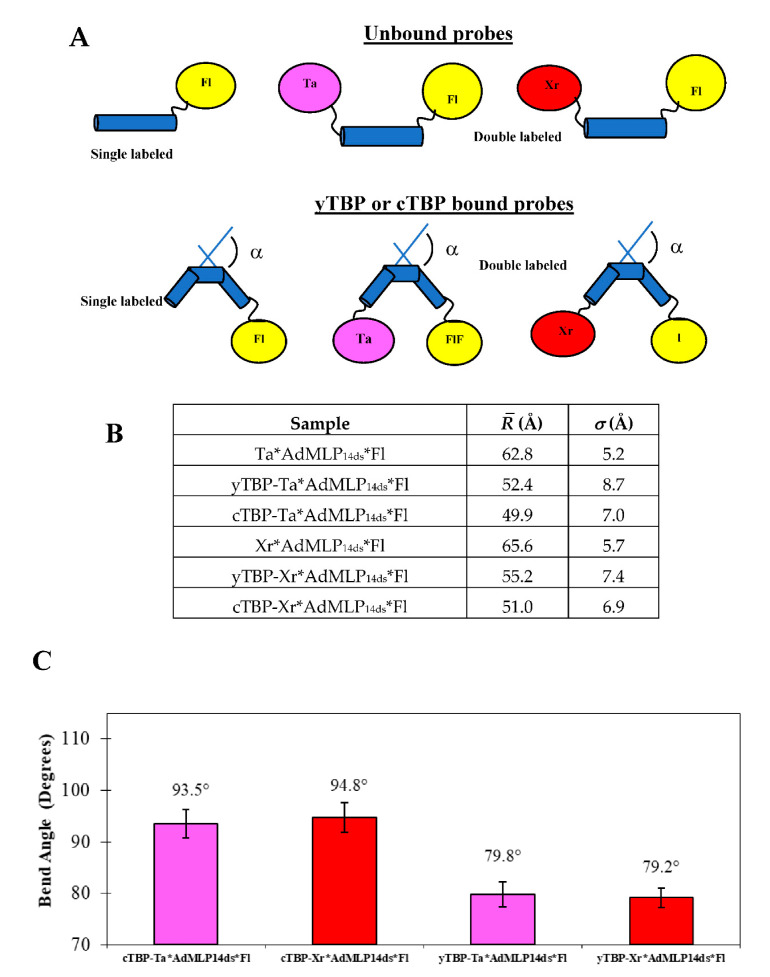
The interdye distances and bend angles of free AdMLP_14ds_ probes and bound to cTBP and yTBP, at 20 °C. (**A**) To calculate the *P(R)* distance distribution, it is required to obtain the lifetime of the single and double-labeled probe as free and bound complexes with TBP proteins. The Xr*AdMLP_14ds_*Fl and Ta*AdMLP_14ds_*Fl probes were straight but after complex formation with yTBP or cTBP, the DNA is bent, which shortens the interdye *R* resulting in more energy transfer from Fl donor toward the acceptor (Ta or Xr). (**B**) The *P(R)* distance distribution is described by the mean distance $\bar{R}$ and the spread *σ* of the unbound probes and the respective complexes formed with yTBP and cTBP. (**C**) The bend angle (*α*) of the TATA distortion for the canonical AdMLP is calculated according to Equation (25), requiring the $\bar{R}$-value and where *L2* was 20.4 Å. The bend angle produced by each protein was independent of the probe used since the values overlapped in error for the Xr*AdMLP_14ds_*Fl and Ta*AdML_P14ds_*Fl complexes formed with yTBP and cTBP, respectively. The bend angles caused by cTBP for both probes were larger than those observed by yTBP since the latter has an N-terminal domain that regulated the binding and shifts the equilibrium to the left, or towards the reactants [40].

**Figure 10 biosensors-10-00180-f010:**
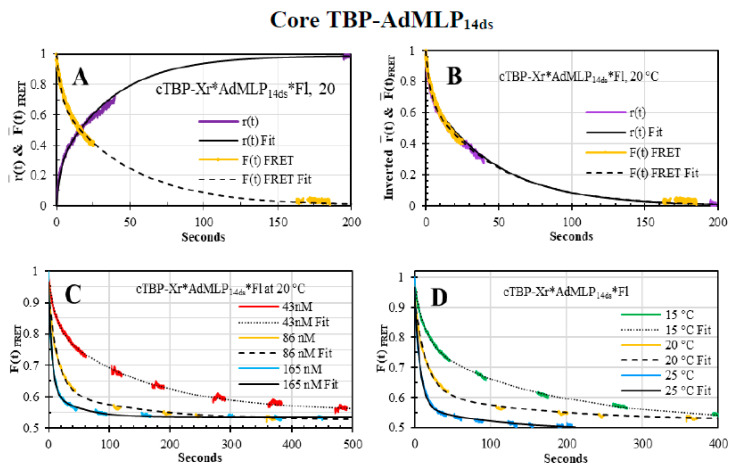
Core TBP-AdMLP_14ds_ association kinetics. (**A**,**B**) The association kinetics of cTBP (260 nM)-Xr*AdMLP_14ds_ (60 nM) and cTBP (86 nM)-Xr*AdMLP_14ds_*Fl (20 nM) were monitored by $\bar{r}\left(t\right)$ (orange) and $\bar{F}{\left(t\right)}_{FRET}$ (purple) sensing modalities, at 20 °C. The $\bar{r}(t$*)* and $\bar{F}{\left(t\right)}_{FRET}$ fits yielded eigenvalues of 0.350 (±0.040) s^−1^ and 0.337 (±0.078) s^−1^, which resulted in *k_on_* values of 1.35 (±0.05) × 10^6^ M^−1^s^−1^ and 1.30 (± 0.10) × 10^6^ M^−1^s^−1^, respectively. Both *k_on_* values are in excellent agreement since the protein/probe ratio is 4.3 and they are in the error range of the reported *k_+_*_1_ of 1.26 (±0.05) × 10^6^ M^−1^s^−1^, at 25 °C [2]. (**C**) The concentration dependence of the cTBP-AdMLP_14ds_ complex was observed by the $\bar{F}{\left(t\right)}_{FRET}$ sensing modality with cTBP at 43 nM (red), 86 nM (orange), and 165 nM (blue) concentrations, at 20 °C, reacting with 20 nM Xr*AdMLP_14ds_*Fl. (**D**) The temperature dependence was also monitored by the $\bar{F}{\left(t\right)}_{FRET}$ traces, at 15 °C (green), 20 °C (orange), and 25 °C (blue) of 86 nM cTBP reacting with 20 nM Xr*AdMLP_14ds_*Fl. The FRET global fits (black lines) of the concentration and temperature dependence traces yielded a two-intermediate reaction mechanism with *k_+_*_1_ values of 9.62 (±0.41) × 10^5^ M^−1^s^−1^, 1.26 (±0.05) × 10^6^ M^−1^s^−1^ and 1.64 (±0.06) × 10^6^ M^−1^s^−1^ at 15 °C, 20 °C and 25 °C, respectively [2].

**Figure 11 biosensors-10-00180-f011:**
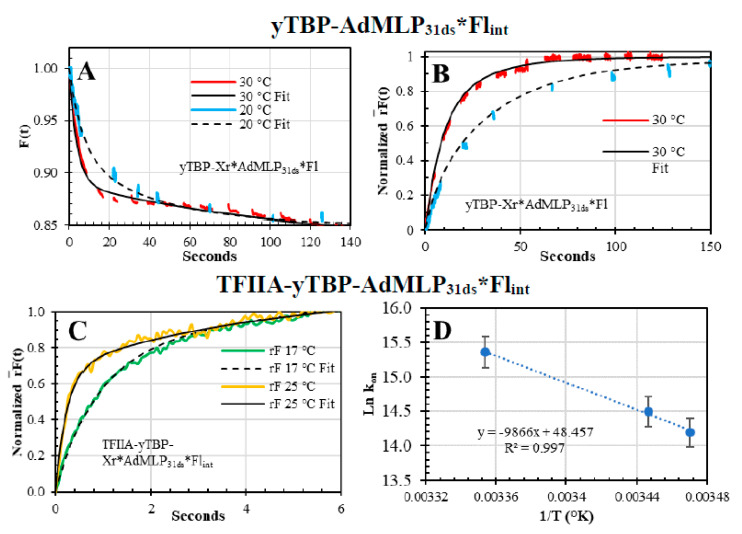
Association kinetic of binary yTBP-AdMLP_31ds_*Fl_int_ and ternary complex TFIIA-yTBP-AdMLP_31ds_*Fl_int_. (**A**) The association traces of yTBP (220 nM) and AdMLP_31ds_*Fl_int_ (20 nM), at 20 °C (blue) and 30 °C (red), were monitored by the *F(t)* sensing modality, yielding parameters of *α*_1_ = 8.0% (±1.0%), *λ*
_1_= 0.15 s^−1^ (±0.010 s^−1^), *α*_2_ = 7.0% (±1.0%), *λ*_2_ = 0.025 s^−1^ (±0.010 s^−1^), baseline = 85.0% ± 2.0%; and *α*_1_ = 9.8% (±0.3%), *λ*_1_ = 0.26 s^−1^ (±0.03 s^−1^), *α*_2_ = 5.2% (±0.3%), *λ*_2_ = 0.0085 s^−1^ (±0.0015 s^−1^), baseline = 85.0% ± 2.0%, respectively. (**B**) The $\bar{rF}\left(t\right)$ association traces of binary yTBP (220 nM)-AdMLP_31ds_*Fl_int_ (20 nM) complex, at 20 °C, yielded the following parameters: *α*_1_ = 27.2% (±11.6%), *λ*_1_= 0.1245 s^−1^ (±0.0133) s^−1^, *α*_2_ = 72.8% (±6.1)%, *λ*_2_ = 0.0221 s^−1^ (±0.0060 s^−1^); and at 30 °C, the parameters were: *α*_1_ = 35.0% (±8.0)%, *λ*_1_ = 0.30 s^−1^ (±0.03 s^−1^), *α*_2_ = 65.0% (±7.0%), *λ*_2_ = 0.925 s^−1^ (±0.06 s^−1^). The *k_on_* values for this longer probe in the yTBP-AdMLP31ds*Fl_int_, at 20 °C and 30 °C, were calculated with the faster *λ*_1_ resulting in values of 5.91 (± 0.46) × 10^5^ M^−1^s^−1^ and 4.20 (± 0.27) × 10^5^ M^−1^s^−1^, respectively, which were in excellent agreement with the respective values of 5.80 (± 0.26) × 10^5^ M^−1^s^−1^ and 4.21 (± 0.19) × 10^5^ M^−1^s^−1^, obtained with the 14-nucleotide probe (yTBP-AdMLP_14ds_*Fl), at the same temperatures, respectively [11]. (**C**) The dual-channel-SF association of yTFIIA (850 nM) and the binary yTBP (220 nM)-AdMLP_31ds_*Fl_int_ (20 nM) complex, at 15 °C (blue), 17 °C (green), and 25 °C (orange), were biphasic with normalized values of *α*_1_ = 55.5% (±9.0%), *λ*_1_ = 1.24 s^−1^ (±0.33 s^−1^), *α*_2_ = 44.5% (±15.0%), *λ*_2_ = 0.18 s^−1^ (±0.05 s^−1^); *α*_1_ = 53.7% (±8.5%), *λ*_1_ = 1.56 s^−1^ (±0.33 s^−1^), *α*_2_ = 39.5 % (±12.7%), *λ*_2_ = 0.20 s^−1^ (±0.06 s^−1^); and *α*_1_ = 63.4% (±0.06%), *λ*_1_ = 3.98 s^−1^ (±1.35 s^−1^), *α*_2_ = 32.6% (±7.6%), *λ*_2_ = 0.24 s^−1^ (±0.01 s^−1^), respectively. The calculated *k_on_* values, at 15 °C, 17 °C and 25 °C, were 1.45 (±0.3) × 10^6^ M^−1^s^−1^, 1.84 (±0.34) × 10^6^ M^−1^s^−1^ and 4.68 (±1.59) × 10^6^ M^−1^s^−1^, respectively. (**D**) The calculated van’t Hoff plot yielded an enthalpy of 19.6 ± 1.6 Kcal/mol for the yTFIIA and yTBP-AdMLP_31ds_*Fl_int_ binding process.

**Table 1 biosensors-10-00180-t001:** Stopped-flow association reactions monitored by *F(t)*, *r(t)*, *rF(t),* and *F(t)_FRET_* sensing modalities.

SF Methodology	Signal	Reaction	Syringe 1 ^a^	Syringe 2
**Magic angle**	***F(t)***	**AV-BcO**	**BcO**	AV
**Polarizers**	***r(t), F(t),*** **& *rF(t)***	**AV-BcO**	**BcO**	AV
		**yTBP-Ta*AdMLP_14ds_*Fl**	5′-**Xr***GGGCTATAAAAGGC***Fl**-3′ 3′-CCCGATATTTTCCG-5′	yTBP
		**cTBP-Xr*AdMLP_14ds_**	5′-Xr*GGGCTATAAAAGGC***Fl**-3′ 3′-CCCGATATTTTCCG-5′	cTBP
**Dual-Channel**	***r(t), F(t),*** **& *rF(t)***	**AV-BcO**	**BcO**	AV
		**yTBP-Ta*AdMLP_14ds_**	5′-**Ta***GGGCTATAAAAGGC-3′ 3′-CCCGATATTTTCCG-5′	yTBP
		**yTBP-AdMLP_14ds_*Fl**	5′-GGGCTATAAAAGGC***Fl**-3′ 3′-CCCGATATTTTCCG-5′	yTBP
		**yTBP-AdMLP_31ds_*Fl_int_**	5′-GCGGGGAATTCCTATAAAAGAA(T-**Fl**)GTGCTGGG-3′ 3′-CGCCCCTTAAGGATATTTTCTTACACGACCC-5′	yTBP
		**yTFIIA-yTBP-AdMLP_31ds_*Fl_int_^b^**	yTBP + 5′-GCGGGGAATTCCTATAAAAGAA(T-**Fl**)GTGCTGGG-3′ 3′-CGCCCCTTAAGGATATTTTCTTACACGACCC-5′	yTFIIA
**FRET**	***F(t)_FRET_***	**yTBP-Xr*AdMLP_14ds_*Fl**	5′-**Xr***GGGCTATAAAAGGC***Fl**-3′ 3′-CCCGATATTTTCCG-5′	yTBP
		**cTBP-Xr*AdMLP_14ds_*Fl**	5′-**Xr***GGGCTATAAAAGGC***Fl**-3′ 3′-CCCGATATTTTCCG-5′	cTBP

^a^ Labeled top strand and complement were preincubated before protein binding. ^b^ The TFIIA-TBP-AdMLP_31ds_*Fl_int_ ternary complex was monitored by pre-forming the yTBP-AdMLP_31ds_*Fl_int_ binary complex in syringe 1.

**Table 2 biosensors-10-00180-t002:** Steady-state anisotropy (*r_ss_*) and quantum yield (*QY*) for unbound probes and complexes.

Sample	*r_ss_* (Free Probe) ^a^	*r_ss_* (Complex) ^b^	*QY* (Free Probe)	*QY* (Complex)
BcO (25 °C)	0.018 ± 0.001 ^c^ 0.025 ± 0.001 ^d^	+AV: 0.180 ± 0.003 ^c^ +AV: 0.177 ± 0.004 ^d^	0.91 ± 0.01	0.68 ± 0.02
BcO (20 °C)	0.025 ± 0.001 ^d^	+AV: 0.185 ± 0.004 ^d^		
BcO (15 °C)	0.054 ± 0.001 ^d^	+AV: 0.176 ± 0.004 ^d^		
BcO (10 °C)	0.055 ± 0.001 ^d^	+AV: 0.202 ± 0.004 ^d^		
Ta*AdMLP_14ds_ (25 °C)	0.164 ± 0.002 ^d^	+yTBP: 0.192 ± 0.010 ^d^	0.20 ± 0.01	0.20 ± 0.01
Ta*AdMLP_14ds_ (20 °C)	0.122 ± 0.002 ^d^	+yTBP: 0.131 ± 0.005 ^d^	0.20 ± 0.01	0.20 ± 0.01
AdMLP_14ds_*Fl (20 °C)	0.071 ± 0.013 ^d^ 0.068 ± 0.008 ^c^	+yTBP: 0.097 ± 0.002 ^d^	0.22 ± 0.01	0.22 ± 0.01
Xr*AdMLP_4ds_ (20 °C)	0.122 ± 0.003 ^d^ 0.122 ± 0.001 ^c^	+cTBP: 0.130 ± 0.002 ^d^	0.10 ± 0.01	0.10 ± 0.01
AdMLP_31ds_*Fl_int_ (20 °C)	0.043 ± 0.004 ^c^ 0.040 ± 0.004 ^d^	+yTBP: 0.201 ± 0.005 ^d^	0.83 ± 0.03	0.71 ± 0.03
yTBP-AdMLP_31ds_*Fl_int_ (17 °C)	0.201 ± 0.005 ^d^	+yTFIIA: 0.260 ± 0.002 ^d^	0.71 ± 0.03	0.82 ± 0.03
yTBP-AdMLP_31ds_*Fl_int_ (25 °C)	0.198 ± 0.005 ^d^	+yTFIIA: 0.217 ± 0.002 ^d^	0.71 ± 0.03	0.72 ± 0.03

^a^ The *r_ss_* values were used to solve *r(t)* and *F(t)* in Equations (7) and (8), since the *r_ss_* values of the free and bound probes corresponded to the association anisotropy traces at initial and endpoint, *r(t = 0),* and *r(t = ∞)*, respectively. The unbound probes have low *r_ss_* values since they are free to rotate; however, the anisotropy increased when the probe is bound in the complex. ^b^ Protein was added to at least 10× excess to reach at least 98% saturation of the fluorescent probe. ^c^ Calculated with polarizers. ^d^ Calculated with the method by Giblin-Parkhurst [30].

**Table 3 biosensors-10-00180-t003:** The eigenvalues *(λ)* and calculated association rate constants *(k_on_)* of the association reaction of BcO (20 nM) and AV (200 nM) at 20 °C, obtained with the $\bar{F}\left(t\right)$ and $\bar{rF}\left(t\right)$ sensing modalities, and acquired by the tree stopped-flow methodologies.

Fluorescence, $\bar{F}\left(t\right)$ ^a^	*λ* (s^−1^)	*k_on_* (×10^−6^ M^−1^s^−1^)	Error (%) ^c^
**Dual channel**	1.186 ± 0.043	5.93 ± 0.22	3.6
**Polarizers**	1.187 ± 0.083	5.94 ± 0.42	7.0
**Magic angle**	1.199 ± 0.099	5.99 ± 0.60	8.3
***rF****(t)*** ^**b**^	***λ*** (**s**^**−1**^)	***k**_**on**_***(×10**^**−6**^**M**^**−1**^**s**^**−1**^)	**Error (%)** ^**d**^
**Dual channel**	1.183 ± 0.023	5.92 ± 0.12	2.0
**Polarizers**	1.198 ± 0.039	5.99 ± 0.20	3.3
**Magic angle**	NA	NA	NA

^a^ The photobleaching was discarded from the reaction model. ^b^ The $\bar{rF}\left(t\right)$ is the product of *r(t)* × *F(t),* which corrects the distortion of the *r(t)* traces by changes in the *QY_i_* [17]. ^c^ The $\bar{F}\left(t\right)$ errors acquired by polarizers and magic angle methodologies were ~2× larger than the errors observed with the dual-channel and magic-angle SF methodology. ^d^ The $\bar{rF}\left(t\right)$ error acquired by polarizers was 1.65× larger than the error observer with the dual-channel methodology.

**Table 4 biosensors-10-00180-t004:** Fitted eigenvalues (*λ*) and association rate constants (*k_on_*) of AV-BcO as a function of concentration and temperature, at pH 8, and under pseudo-first-order conditions. ^a.^

	200 nM		260 nM		520 nM		1040 nM		
***λ*** **(s^−1^)**	$\bar{F}\left(t\right)$	$\bar{rF}\left(t\right)$	$\bar{F}\left(t\right)$	$\bar{rF}\left(t\right)$	$\bar{F}\left(t\right)$	$\bar{rF}\left(t\right)$	$\bar{F}\left(t\right)$	$\bar{rF}\left(t\right)$	
**10 °C**	NA	NA	0.629 ± 0.042	0.690 ± 0.035	1.355 ± 0.047	1.372 ± 0.003	2.825 ± 0.107	2.620 ± 0.012	
**15 °C**	NA	NA	1.058 ± 0.016	1.035 ± 0.005	1.840 ± 0.055	1.840 ± 0.009	4.000 ± 0.108	4.200 ± 0.365	
**20 °C**	1.186 ± 0.043	1.183 ± 0.023	1.491 ± 0.027	1.543 ± 0.014	3.045 ± 0.085	3.024 ± 0.030	6.209 ± 0.571	6.357 ± 0.058	
**25 °C**	NA	NA	2.465 ± 0.014	2.473 ± 0.018	4.920 ± 0.093	4.920 ± 0.034	10.031 ± 0.401	9.810 ± 0.091	
***k_on_*** **×** **10^−6^ M^−1^s^−1^**	$\bar{F}\left(t\right)$	$\bar{rF}\left(t\right)$	$\bar{F}\left(t\right)$	$\bar{rF}\left(t\right)$	$\bar{F}\left(t\right)$	$\bar{rF}\left(t\right)$	$\bar{F}\left(t\right)$	$\bar{rF}\left(t\right)$	***k_on_*** **(Average)**
**10 °C**	NA	NA	2.419 ± 0.162	2.655 ± 0.133	2.606 ± 0.091	2.638 ± 0.006	2.520 ± 0.010	2.716 ± 0.012	2.592 ± 0.107
**15 °C**	NA	NA	4.069 ± 0.061	3.980 ± 0.021	3.539 ± 0.106	3.539 ± 0.018	3.846 ± 0.104	4.038 ± 0.351	3.835 ± 0.242
**20 °C**	5.931 ± 0.216	5.931 ± 0.117	5.733 ± 0.103	5.935 ± 0.054	5.855 ± 0.164	5.815 ± 0.058	5.970 ± 0.549	6.113 ± 0.056	5.904 ± 0.133
**25 °C**	NA	NA	9.479 ± 0.052	9.513 ± 0.068	9.461 ± 0.180	9.461 ± 0.066	9.646 ± 0.386	9.433 ± 0.088	9.499 ± 0.077

^a^ The association reactions were acquired with BcO (20 nM) binding to AV at 200 nM, 260 nM, 520 nM, and 1040 nM concentrations from 10 °C to 25 °C at pH 8. The normalized $\bar{F}\left(t\right)$ and $\bar{rF}\left(t\right)$ sensing modalities yielded equivalent bimolecular rate constant (*k_on_*) for the BcO binding to the AV at each temperature.

**Table 5 biosensors-10-00180-t005:** Time-resolved FRET of the free duplexes and TBP bound to AdMLP_14ds_*Fl, Xr*AdMLP_14ds_*Fl, and Ta*AdMLP_14ds_*Fl, at 20 °C. All the decays were best described by a bi-exponential decay model according to the statistical parameters *χ^2^*, Durbin-Watson (*DW*), and *Z* run (Equation (21)).

Sample	*χ*	*DW*	*Z*	*α*	*τ* *(ns)*	*α*	*τ* *(ns)*	*φ* *(ns) ^a^*	*φ* *(ns)*	*α_i_* *τ_i_ (ns) ^b^*
**AdMLP_14ds_*Fl**	0.985 ± 0.030	1.965 ± 0.135	0.069 ± 0.400	0.340 ± 0.033	0.852 ± 0.098	0.660 ± 0.033	3.729 ± 0.043	0.105 ± 0.012	0.895 ± 0.012	2.749 ± 0.132
**Xr*AdMLP_14ds_*Fl**	1.008 ± 0.030	2.027 ± 0.115	−0.075 ± 0.352	0.494 ± 0.033	0.645 ± 0.097	0.506 ± 0.033	2.493 ± 0.119	0.201 ± 0.022	0.799 ± 0.022	1.575 ± 0.066
**Ta*AdMLP_14ds_*Fl**	0.984 ± 0.040	1.942 ± 0.067	−0.021 ± 0.170	0.520 ± 0.014	0.607 ± 0.034	0.480 ± 0.014	2.230 ± 0.037	0.279 ± 0.049	0.949 ± 0.173	1.385 ± 0.042
**cTBP-AdMLP_14ds_*Fl**	1.009 ± 0.022	1.987 ± 0.132	−0.006 ± 0.404	0.315 ± 0.013	0.826 ± 0.123	0.685 ± 0.013	3.823 ± 0.084	0.090 ± 0.012	0.910 ± 0.012	2.878 ± 0.096
**yTBP-AdMLP_14ds_*Fl**	1.016 ± 0.029	1.916 ± 0.086	−0.050 ± 0.349	0.330 ± 0.018	0.852 ± 0.060	0.670 ± 0.018	3.998 ± 0.127	0.095 ± 0.005	0.905 ± 0.005	2.959 ± 0.149
**cTBP-Xr*AdMLP_14ds_*Fl**	1.018 ± 0.026	2.017 ± 0.157	0.160 ± 0.313	0.715 ± 0.036	0.428 ± 0.043	0.285 ± 0.036	1.763 ± 0.112	0.381 ± 0.048	0.619 ± 0.048	0.805 ± 0.034
**yTBP-Xr*AdMLP_14ds_*Fl**	1.016 ± 0.010	1.916 ± 0.045	−0.185 ± 0.114	0.668 ± 0.013	0.529 ± 0.035	0.332 ± 0.013	2.077 ± 0.047	0.339 ± 0.010	0.661 ± 0.010	1.042 ± 0.040
**cTBP-Ta*AdMLP_14ds_*Fl**	1.011 ± 0.024	1.972 ± 0.142	0.388 ± 0.234	0.772 ± 0.012	0.434 ± 0.010	0.228 ± 0.012	1.601 ± 0.022	0.335 ± 0.010	0.364 ± 0.014	0.699 ± 0.010
**yTBP-Ta*AdMLP_14ds_*Fl**	0.974 ± 0.058	1.902 ± 0.106	0.299 ± 0.344	0.806 ± 0.008	0.564 ± 0.032	0.194 ± 0.008	2.264 ± 0.068	0.509 ± 0.007	0.491 ± 0.007	0.894 ± 0.046

^a^ The *φ* parameters are the normalized contribution of each phase in nanoseconds (ns). ^b^ The area under the deconvoluted decay is described by Σ*α_i_τ_i_*.

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
