# Peer review of "Dual-Channel Stopped-Flow Apparatus for Simultaneous Fluorescence, Anisotropy, and FRET Kinetic Data Acquisition for Binary and Ternary Biological Complexes"

_biosensors, 2020, doi:10.3390/bios10110180_

Round 1

Reviewer 1 Report

This paper presents a dual-channel stopped-flow apparatus for fluorescence, anisotropy, and FRET kinetic data acquisition, simultaneously. And can be used for studying the reactions of binary and ternary biological complexes. Here, I have some comments and suggestions:

1.The position labels of the figures are not the same. For instance, in fig. 5, A is at the left-up corner and B is at the left down corner. Their position should be the same. There are many figures have the same problem.

  1. Fig.2, the words and figures are overlapped.

  1. The format of Fig. 9 is not correct.

  1. in Line 656, the title of the figure shouldn’t be put there.

  1. The conclusion is too long. The authors should summarize the results but not to describe all of them again.

  1. The introduction should be improved. For example, the expression way of fig. 1 is confusing. And also, too many details are in the introduction. However, I could understand clearly what the authors want to express.

  1. For the results part, the interpretation is also not clear. The reader would be confusing what the authors want to express.

Reviewer 2 Report

The present paper presents a new design to analyze the fluorescent and anisotropy signals using a modified L-Type stopped flow instrument allowing for the simultaneous detection of the parallel and vertical components by means of two independent channels. In this regard, the paper is of interest. Aparte to present the design of the instrument, authors make an extensive validation by means of different binding and biochemical reactions of biological interest, confirming their initial hypothesis. for this reason, the paper deserves publication. Several minor points should be corrected prior publication:

  1. Numbering of Subsection 2.4 and 2.5 within general section 2.3 is confusing. Please renumbering.
  2. In page 12 different spacing and font sizes coexist. Please, correct.
  3. In page 14. Explanations of G factors obtained and the conditions in which there were obtained/calculated are confusing. Please rewrite.
  4. Figure 11D: This Figure and the corresponding fitting is meaningless. A straight line cannot be defined with only two experimental points.

Round 2

Reviewer 1 Report

I think it improves a lot.